# Environmental memory boosts group formation of clueless individuals

Cristóvão S. Dias[1,2], Manish Trivedi[3], Giovanni Volpe [4] ✉,
Nuno A. M. Araújo [1,2] ✉ & Giorgio Volpe [3] ✉

The formation of groups of interacting individuals improves performance and fitness in many decentralised systems, from micro-organisms to social insects, from robotic swarms to artificial intelligence algorithms. Often, group formation and high-level coordination in these systems emerge from individuals with limited information-processing capabilities implementing low-level rules of communication to signal to each other. Here, we show that, even in a community of clueless individuals incapable of processing information and communicating, a dynamic environment can coordinate group formation by transiently storing memory of the earlier passage of individuals. Our results identify a new mechanism of indirect coordination via shared memory that is primarily promoted and reinforced by dynamic environmental factors, thus overshadowing the need for any form of explicit signalling between individuals. We expect this pathway to group formation to be relevant for understanding and controlling self-organisation and collective decision making in both living and artificial active matter in real-life environments.

Strength in numbers is more than an idiomatic expression. Many living systems form groups to improve their fitness, optimise the use and allocation of resources, and reach consensus[1]. Examples emerge at all length scales, from bacterial quorum sensing and biofilm formation[2] to social insects[3], from animal groups[4] to human crowds[5]. Artificial active matter systems, such as active colloids[6] and robotic swarms[7], provide controllable systems to pinpoint the essential principles behind the emergence of these collective behaviours in living systems[8,9]. For example, active colloids have been employed to demonstrate motility-induced phase separation[10,11] as well as the spontaneous formation of living crystals resembling animal group formation[8,12]. Complex dynamic collective patterns, such as colloidal swarms, flocks and swirls, have also been demonstrated by introducing controllable attractive, repulsive or aligning interactions among individuals by particle design[13,14], by defining appropriate confining potentials[15] or by modulating particles' propulsion with external feedback loops[9,16]. In recent years, a few active particles in crowded environments of passive

colloids have also been employed to modulate the energy landscape of the passive phase with an emphasis on controlling the assembly of soft materials[17–25].

Whether living or artificial, decentralised systems are characterised by high-level coordination and collective behaviours, which emerge from individuals with limited information-processing capabilities responding to low-level rules of engagement[26]. In particular, stigmergy is a form of indirect communication between individuals by means of the environment, either mediated by physical modifications (sematectonic stigmergy) or by a signalling mechanism via deposition of markers (marker-based stigmergy) which shape a shared environmental memory[27]. This strategy underpins the emergence of coordination and collective decision-making in many natural decentralised systems, from micro-organisms[2] to social insects[3]. For example, trailing stalk cells guided by chemo-attractants through tissue establish the vascular lumen in sprout angiogenesis[28]; bacteria[29], amoebas[30] and ants[31] can solve physical mazes by tracking chemical scents and

[1]Departamento de Física, Faculdade de Ciências, Universidade de Lisboa, 1749-016 Lisboa, Portugal. [2]Centro de Física Teórica e Computacional, Faculdade de Ciências, Universidade de Lisboa, 1749-016 Lisboa, Portugal. [3]Department of Chemistry, University College London, 20 Gordon Street, WC1H 0AJ London, UK. [4]Department of Physics, University of Gothenburg, Origovägen 6B, SE-412 96 Gothenburg, Sweden. ✉e-mail: giovanni.volpe@physics.gu.se; nmaraujo@fc.ul.pt; g.volpe@ucl.ac.uk

forming optimal paths; mutual anticipation and avoidance in crowds lead to lane formation and stabilisation[32]. The concept of stigmergy has also found widespread use in technological and engineering applications, from robotic swarms[33] to artificial intelligence algorithms[7], to, recently, active colloids[34]. In these systems, it is usually assumed that individuals possess a minimal level of low-level communication and signal processing capabilities, which leads to the emergence of shared environmental memory and, eventually, high-level group dynamics[27].

Here, we demonstrate that, even in a community of clueless self-motile individuals (i.e., incapable of directly signalling to each other or processing information), avoidance of a dense population of non-fixed obstacles is sufficient to lead to the emergence of stigmergy when the dynamic environment can transiently store memory of the earlier passage of individuals. Counterintuitively, we find that, while the motion of the individuals is hampered by increasing levels of crowding, the spatial correlations created and stored in the otherwise passive environment after their passage feed back on the motion of other individuals to boost aggregation rates and, consequently, group formation.

## Results

### Group formation experiments

As paradigmatic self-motile individuals, we employ Janus silica ($SiO_2$) colloids (diameter $d = 4.77 \pm 0.20$ μm) half-coated with a thin layer of carbon ($\approx 60$ nm) (Methods). When suspended in a critical binary mixture of water and 2,6-lutidine (0.286 mass fraction of lutidine) below its lower critical temperature ($T_c \approx 307$ K), these colloids undergo Brownian diffusion[35]. Upon exposure to laser illumination ($\lambda = 532$ nm, $I \approx 2.5$ μW μm$^{-2}$) (Methods), light absorption at the carbon cap simultaneously propels the Janus particles in the field of view with the more hydrophobic carbon-coated side at the front at a speed of $v \approx 1.9$ μm s$^{-1}$ due to local heating and demixing of the critical mixture around the cap[35]. Our experiments are in the Stokes regime (Reynolds numbers, Re $\approx 10^{-5} \ll 1$) and inertial effects, including those of the fluid[36,37], can be safely neglected. Because of their colloidal nature, these self-motile individuals are clueless in the sense that they have no sensing or information-processing capabilities and interact with each other through simple physical interaction rules, such as steric and short-range attractive interactions[8,10,38]. Boundaries can also influence their motion with aligning interactions[39,40].

To study their interplay with a dynamic environment of non-static obstacles, we prepare quasi-two-dimensional samples of Janus particles mixed with dispersions of equally sized freely diffusing silica ($SiO_2$) colloids at different densities ($0 \le \rho_p \le 75\%$, defined as fractional surface coverage), where the Janus particles only represent a small portion ($0.5\% \le \rho_a \le 1.6\%$, also defined as fractional surface coverage) (Methods). The two example time sequences of active particles ($\rho_a = 1.1\%$) moving in a crowded environment ($\rho_p = 37.5\%$) in Fig. 1 and Supplementary Fig. 1 show how the changes introduced in the passive phase by the active colloids produce spatial correlations in the environment in the form of open transient paths. These paths feedback on the motion of the active particles, eventually leading to group formation (here defined as the formation of a cluster of at least three particles separated by at most $0.1d$ from another particle and surviving for at least one frame). While moving forward, individual Janus particles need to physically dig their own path against the surrounding background of passive colloids (Fig. 1a and Supplementary Fig. 1a). Although the presence of voids in the background of passive particles can simplify this task at times, their overall motility reduces for increasing values of $\rho_p$ as exemplified by the mean square displacements (MSDs) in Supplementary Fig. 2. Unless pushed by a Janus particle, the motion dynamics of the passive particles remain diffusive at the edges of the transient paths (Fig. 1a and Supplementary Fig. 1a).

Interestingly, before closing due to the Brownian motion of the passive colloids, these paths appear to be reused by other active colloids, which favour reusing these preformed paths of lower resistance from either end rather than digging their owns (Fig. 1b and Supplementary Fig. 1b). A form of stigmergy (consistent with the definition of sematectonic stigmergy[27]) between the active particles is then established thanks to their passive counterparts, where the transient paths opened by the active colloids in their surroundings become a shared

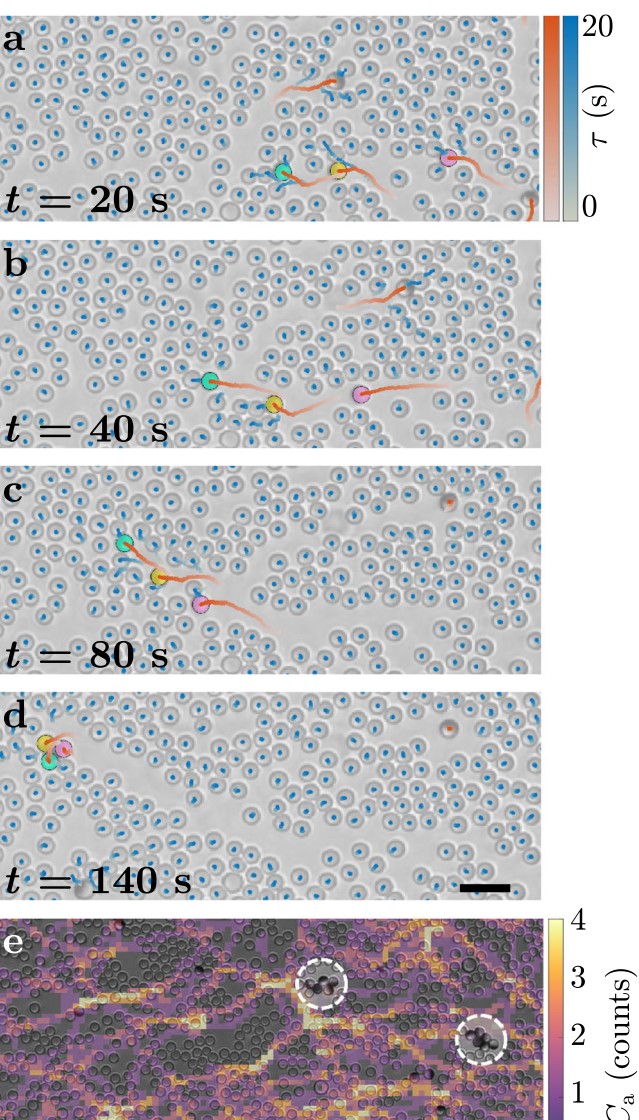

**Fig. 1 | Group formation of active colloids mediated by environmental memory.** **a**–**d** Time sequence showing (**a**) a light-activated Janus particle (cyan) forming a path in a crowded environment of $SiO_2$ passive particles (densities: $\rho_a = 1.1\%$ for active particles and $\rho_p = 37.5\%$ for passive particles), **b**, **c** which is then reused by nearby Janus particles (yellow and magenta) leading to (**d**) the formation of a group (here, a three-particle cluster). In each image, 20-s-long trajectories are shown for both active (red colour scale) and passive (blue colour scale) particles; $t$ represents the time of each frame and $\tau$ the time along each trajectory. Scale bar: 10 μm. **e** Counts $C_a$ (represented as a heatmap) of individual Janus particles that have transited on a pixel during a 16-min acquisition in a similar crowded environment as in **a**–**d**. The heatmap is overlaid to the final frame showing how path generation and reuse (bright lines) correlate to group formation and cohesion of active units in time (white dashed circles). The heatmap was obtained from a sample with $\rho_a = 1.1\%$ and $\rho_p = 37.5\%$ and an occupied pixel was only accounted for once for each particle. Scale bar: 25 μm.

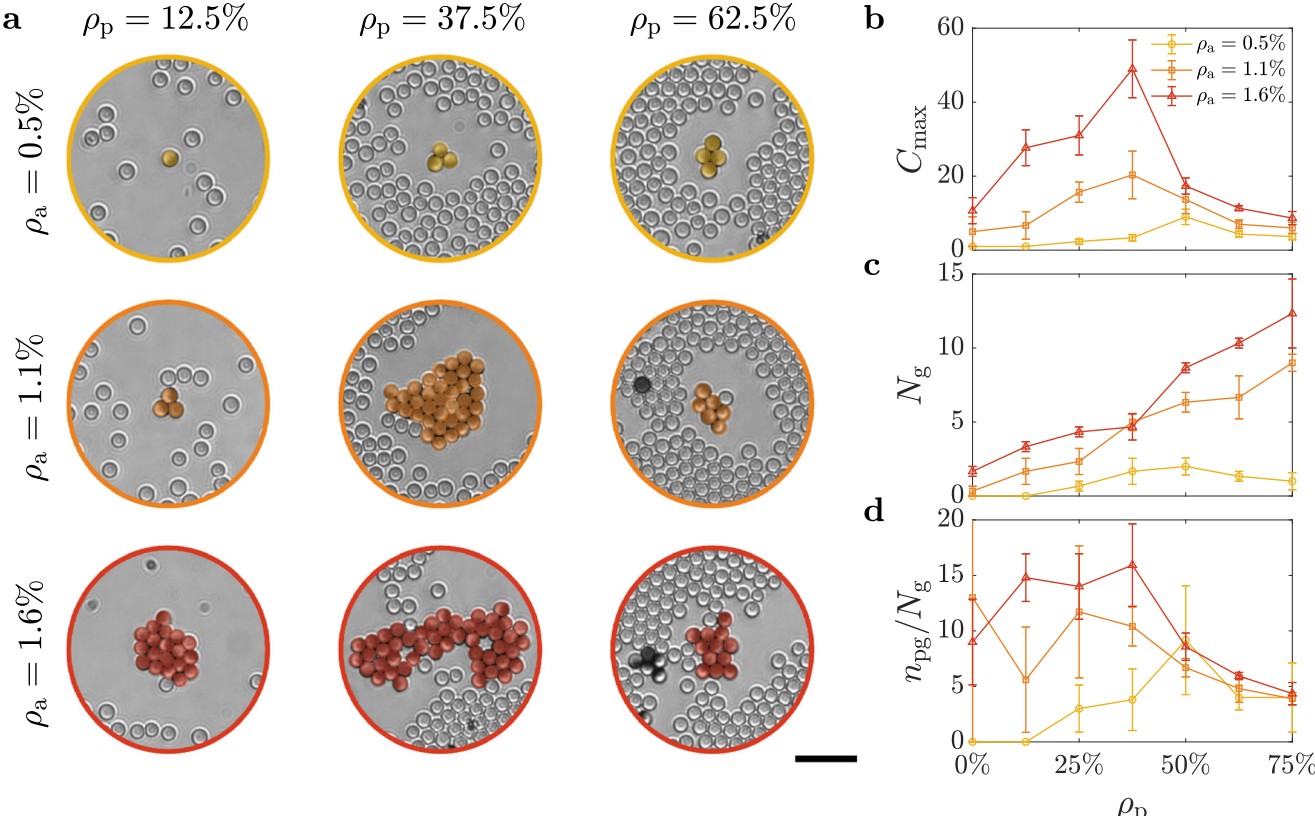

**Fig. 2 | Non-monotonic size dependence of group formation on environmental crowding.** Groups (here defined as all clusters formed by at least 3 individuals) formed after 25 min as a function of the initial densities of active ($\rho_a$) and passive ($\rho_p$) particles. **a** Example images of the largest clusters of active colloids formed at different $\rho_a$ and $\rho_p$. Independent of $\rho_a$, these images show how the largest sizes are obtained at intermediate values of $\rho_p$. Scale bar: 20 μm. **b** This visual trend is confirmed by the non-monotonic dependence of the average size $C_{max}$ of the largest clusters (measured as number of active particles) as a function of $\rho_p$ for different values of $\rho_a$. **c** Differently from $C_{max}$, the total number of groups $N_g$ tends to monotonically increase with $\rho_p$. The error bars in **b** and **c** represent one standard error around the average values obtained from triplicates. **d** Average number of active colloids in a group after 25 min calculated as the ratio between the number of particles in a group $n_{pg}$ and the number of groups $N_g$. The error bars in **d** are obtained by propagating the standard errors on $n_{pg}$ and $N_g$. Source data are provided as a Source Data file.

environmental memory for their peers, which generates a feedback that reinforces their trailing behaviour and, eventually, leads to group formation. Indeed, over time, the trailing Janus colloids catch up with the front particles (Fig. 1c and Supplementary Fig. 1c) to form a small cluster (Fig. 1d and Supplementary Fig. 1d). Once formed, these clusters then grow to larger sizes due to the continuous addition of new active particles to the group through a network of similar transient open paths that form and evolve over time (Fig. 1e). As in other cases of sematectonic stigmergy[27], therefore, the modifications introduced by the active particles in their physical environment (here, the formation of a path in the background of passive colloids) act as asynchronous cues directing the next steps of other particles (here, changing their direction of motion) in a way that provides a direct contribution to the task (here, the carving, reuse and stabilisation of the paths) and facilitates the emergence of population-level coordination (here, group formation).

Figure 2a shows examples of the largest groups obtained after 25 min of experimental time for different values of $\rho_a$ and $\rho_p$. Independently of $\rho_a$, the largest groups appear to form for intermediate values of $\rho_p$, where the crowding is sufficient to create a shared memory in the environment in the form of reusable transient paths, but not dense enough to completely hamper the motility of the individual Janus particles. To quantify this observation, we calculated the size of the largest cluster $C_{max}$ (Fig. 2b), the total number of groups $N_g$ (Fig. 2c) and the average number of particles per group (Fig. 2d) for different values of $\rho_a$ as a function of $\rho_p$. At $\rho_p = 0$, no group forms at a

low density of active particles ($\rho_a = 0.5\%$) as encounters are sparse, while the formation of a very few groups (up to ≈2) becomes increasingly more likely for higher values of $\rho_a$, as chances for encounter increase with the number of available individuals. Increasing $\rho_p$ to intermediate values leads to the formation of more groups on average (for example, up to ≈5 at 37.5%, Fig. 2c). The average size of these groups also shows a tendency to increase for a given $\rho_a$ until a peak is reached around intermediate values of $\rho_p$ (Fig. 2d). Both the exact position of this peak and its width depend on $\rho_a$: the peak is higher, broader, and shifted towards lower values of $\rho_p$, the higher $\rho_a$ (Fig. 2d). This is also reflected by the trend observed for $C_{max}$ (Fig. 2b). Indeed, as chances for encounter increase with the number of available active particles, individuals become more efficient at creating and reusing correlations in their environment through the transient paths, thus lowering the density threshold needed for the passive phase to promote group formation with increasing $\rho_a$. At the peak, a larger cluster tends to emerge (Fig. 2b) that can contain up to ≈67% of the particles in a group ($n_{pg}$) due to the shared environmental memory from the path reuse highlighted in Fig. 1 and Supplementary Fig. 1. This effect is more prominent the higher the value of $\rho_a$. The path reuse by the Janus particles can be quantified through the path revival function $1 - C_{aa}(\tau)$, where $C_{aa}(\tau)$ is the cumulative probability that a region crossed by an active particle will be crossed by another particle within a lag time $\tau$ (Methods). If we consider the particles' velocities to be Poisson distributed when a path is chosen, then this function should follow an exponential distribution for persistent particles of the form

$1 - C_{aa}(\tau) = \exp(-\tau/\tau_{\rho_p})^{41}$, where $\tau_{\rho_p}$ is the effective path revival lifetime, which we fit from the data (Supplementary Fig. 3). The shorter $\tau_{\rho_p}$, the faster $1 - C_{aa}(\tau)$ decays (i.e., the faster $C_{aa}(\tau)$ increases to one), the sooner a region explored by a particle will be crossed by another particle, thus indicating a higher likelihood that a previously opened path will be reused by other active particles. Supplementary Fig. 3a shows how, in our experiments, after initial comparable trends at lower values of $\rho_p$ ($\rho_p \leq 25\%$), the decay of the path revival function becomes faster starting from intermediate values of $\rho_p$ (quantified by an approximately factor-two reduction of the path revival lifetime $\tau_{\rho_p}$ in Supplementary Fig. 3b), thus indicating a higher likelihood of reusing previously explored regions. This change of the path revival lifetime with $\rho_p$ is unexpected and can only by justified by the emergence of a shared environmental memory due to the reuse of pre-existing paths. In fact, if we consider collisions between persistent particles whose velocities are Poisson distributed[41], we would expect the lifetime of the path revival function $\tau_{\rho_p}$ to increase with $\rho_p$ as the particles' effective velocity decreases due to the collisions with the passive particles (as confirmed by the MSDs in Supplementary Fig. 2). Eventually, further increasing $\rho_p$ has a dramatic effect on group formation, as the reduced motility for the active colloids due to the resistance offered by the passive particles (Supplementary Fig. 2) induces a more intuitive behaviour where group formation and cohesion are drastically hampered by the crowded environment and any reduction of the path revival lifetime with respect to the case at $\rho_p = 0\%$ (Supplementary Fig. 3) is now driven by the active particles being more localised in space due to crowding rather than the presence of longer-range correlations in the form of transient paths. For a given $\rho_a$, the ever-increasing number of groups $N_g$ with $\rho_p$ (Fig. 2c) translates now into smaller groups of more homogeneous sizes and more localised in space, which, differently from $N_g$, show a milder dependence on the initial value of $\rho_a$ (Fig. 2b, d).

## Mechanism behind stigmergy of clueless active colloids

To shed light on the physical mechanism behind the reuse of paths in Fig. 1 and Supplementary Fig. 1, Fig. 3a, b shows a close-up of the motion of a Janus particle in the background of passive particles. At $t = 24$ s (Fig. 3a), the particle approaches a block of relatively packed obstacles. On approaching one of them, it turns clockwise towards a void in the structure formed by the passive colloids instead of pushing ahead. At $t = 34$ s, the Janus particle approaches two obstacles head-on via their middle, thus pushing them out of the way and continuing its journey towards a pre-existing path (Fig. 3b). Interestingly, the particle does not align with this path until it encounters a new block of relatively packed obstacles ($t = 47$ s). At this point, a new reorientation event turns it counter-clockwise, thus aligning the particle's motion to the open path ($t = 58$ s) and allowing its reuse. To interpret these reorientation events, we need to consider how the presence of the obstacles affects the particle's self-propulsion mechanism. In their absence, light absorption at the particle's carbon cap induces the local demixing of the water–2,6-lutidine critical mixture, leading to the formation of a lutidine-rich droplet around the more hydrophobic carbon side and of a smaller water-rich droplet around the more hydrophilic silica side[35,42]. The formation of these droplets, which propels the particle[35,42], is mostly symmetric around the axis defined by the direction of the particle's motion, and the particle's reorientation in a viscous medium is driven by its intrinsic rotational diffusion dynamics[35,42]. If obstacles are present on one side, the local demixing is no longer symmetric around the direction of the particle's motion. This asymmetry induces a deterministic aligning torque that reorients the particle towards the more demixed side, i.e., eventually away from the obstacles[42]. As can be seen in Fig. 3a, b, this reorientation (and, hence, the presence of the torque) depends on the local configuration of obstacles and continues until the particle is aligned in a direction where the demixing around the caps is symmetric again and the

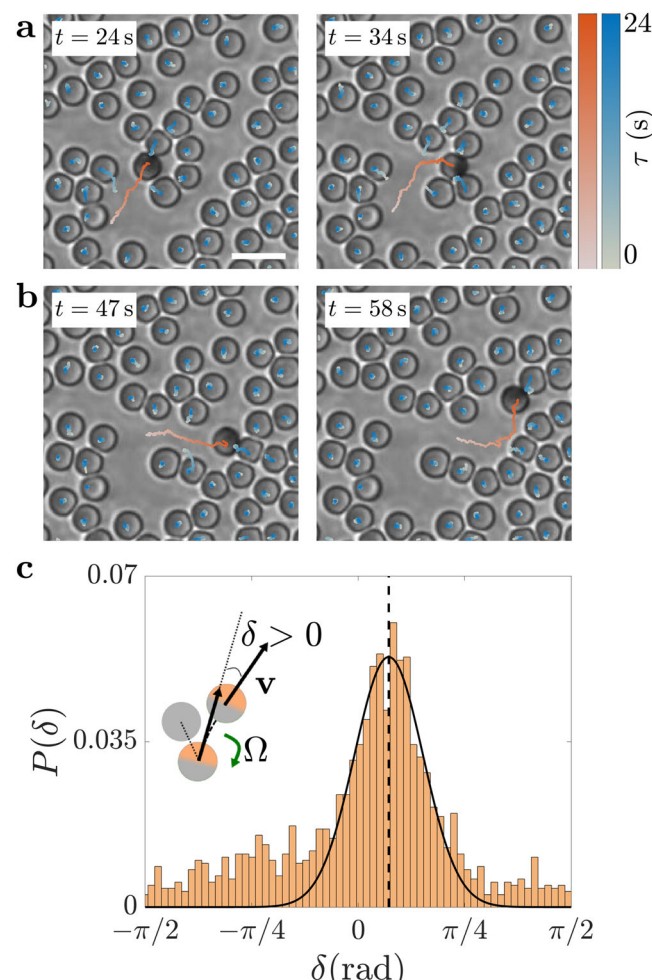

**Fig. 3 | Aligning torque as the mechanism for path reuse. a, b** Time sequence of a light-activated Janus particle experiencing a torque that aligns it to openings in a crowded environment at a density of $SiO_2$ passive particles of $\rho_p = 37.5\%$. The effect of this torque on the particle's trajectory is evident in the frames at **a** $t = 34$ s and **b** $t = 58$ s, where the particle has turned clockwise (towards a void in the structure formed by the passive colloids) and counter-clockwise (towards an open path), respectively. In each image, 24-s-long trajectories are shown for both active (red colour scale) and passive (blue colour scale) particles; $t$ represents the time of each frame and $\tau$ the time along each trajectory. Scale bar: 10 μm. **c** Probability $P(\delta)$ that a Janus particle moving at a velocity **v** is deflected by an angle $\delta$ from its direction of approach to an obstacle due to a torque $\Omega$ (inset). The deflection angle $\delta$ is defined between the active particle's direction of approach to the obstacle (at a distance between the two particles' surfaces equal to one diameter $d$) and its direction of motion after having passed it (after travelling a $2d$ distance); we calculated $\delta$ at low density of passive particles ($\rho_p = 12.5\%$) to primarily consider interactions with single obstacles for active particles' approaches within a narrow angular cone ($\pm\pi/8$ excluding near head-on approaches, which contribute symmetrically around $\delta = 0$). Positive $\delta$ values indicate deflection away from the obstacles (as shown in the inset), while negative $\delta$ values indicate initial deflections towards the obstacles. The solid line is a Gaussian fit (centred at $\delta \approx 0.23$ rad, dashed vertical line) around the position of the peak. Source data are provided as a Source Data file.

aligning torque vanishes (for example, along an open path). To quantify the previous observation, we have calculated the angle $\delta$ at which the Janus colloids are deflected by their interaction with the obstacles (Fig. 3c). Positive and negative $\delta$ values indicate that the particles are deflected initially away from or towards the obstacles, respectively. When steering away from the obstacles (positive $\delta$), we can then expect the particles to immediately align to openings in the background of passive particles, while deflections towards the obstacles

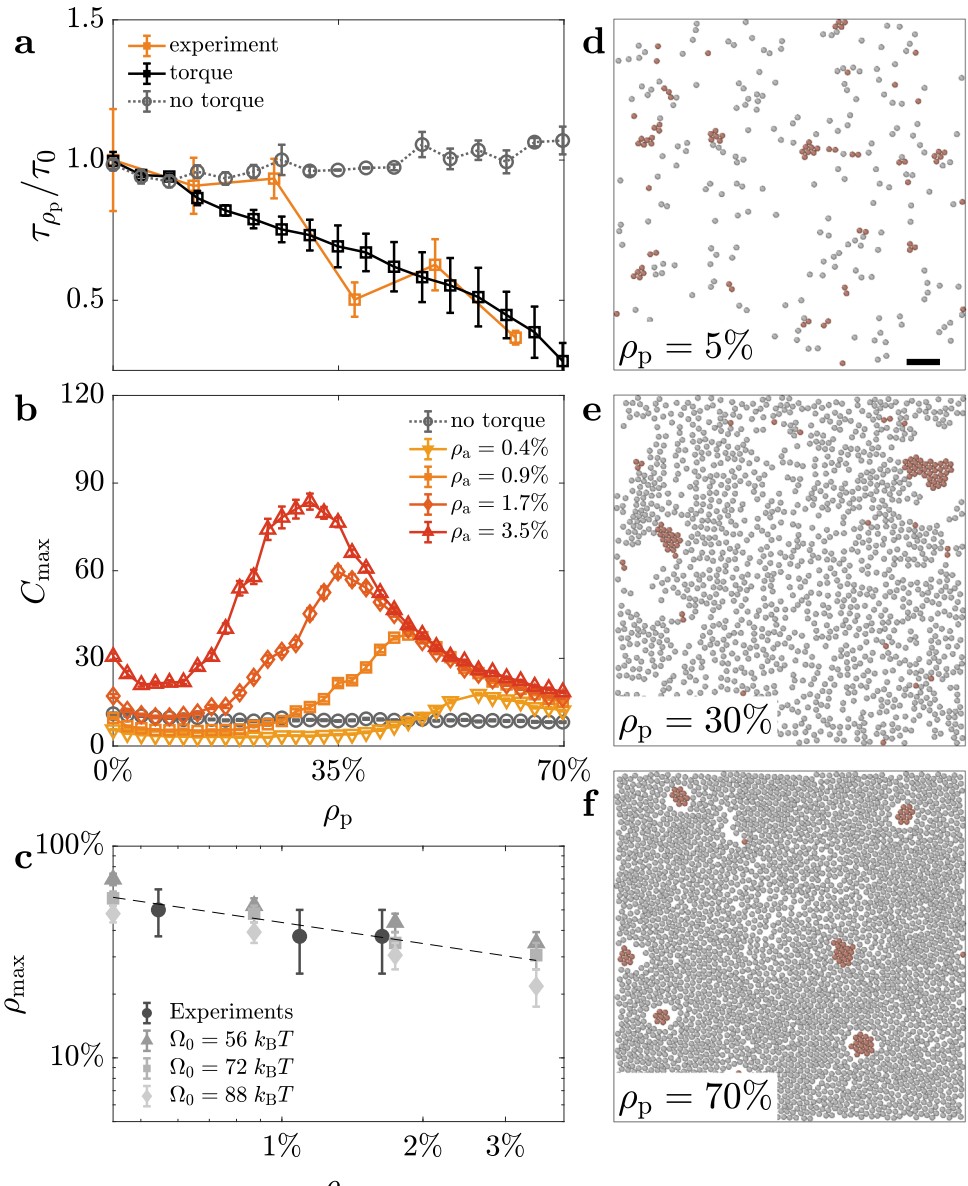

**Fig. 4 | Importance of aligning interactions for stigmergy of active particles.** **a** Path revival lifetime $\tau_{\rho_p}$ as a function of the density of passive particles $\rho_p$ in experiments for a density of active particles $\rho_a = 1.1\%$ (orange solid line) and in simulations with (black solid line) and without (grey dotted line) an effective aligning torque steering the active particles away from the passive ones. The torque $\Omega$ (with strength $\Omega_0 = 72 \pm 16\,k_B T$) depends on the local configuration of obstacles (Methods) and is necessary to reproduce the overall experimental trend. The data from Supplementary Fig. 3 are here normalised to $\tau_0$, the path revival lifetime at $\rho_p = 0$. **b** Simulated average size $C_{max}$ of the largest group for different $\rho_a$ as a function of $\rho_p$ in the presence (coloured solid lines) and absence (grey dotted line) of an aligning torque, showing that the torque is key to the appearance of a peak at intermediate values of $\rho_p$ (as in Fig. 2b). **c** The value of $\rho_p$ at the peak position ($\rho_{max}$) as a function of $\rho_a$ provides a second estimate for the experimental torque strength ($\Omega_0 = 72 \pm 16\,k_B T$). Its monotonously decreasing trend is visualised by a dashed line as a guide for the eyes. Each experimental data point in **a** and **c** is obtained as an average from three videos at the corresponding values of $\rho_a$ and $\rho_p$ and simulations in **a**–**c** are averages over 100 numerical experiments per value of $\rho_a$ and $\rho_p$. All error bars represent one standard error around the average values. **d**–**f** Example snapshots from simulations showing group formation of active particles (red, $\rho_a = 1.1\%$) at different densities of passive particles: **d** $\rho_p = 5\%$, **e** $\rho_p = 30\%$ and **f** $\rho_p = 70\%$. Scale bar: 40 μm. Source data are provided as a Source Data file.

(negative $\delta$) would initially increase chances of collisions with the obstacles until the particle digs a new path or aligns to an existing one. The distribution of angles in Fig. 3c is peaked at positive $\delta$ values (as highlighted by the Gaussian fit centred at $\delta \approx 0.23$ rad), thus quantitatively confirming the stronger tendency for active particles to avoid obstacles and align to open paths already on approach.

To gain a microscopic understanding of the non-monotonic dynamics of group formation in Fig. 2b, we can therefore consider a simple particle-based model that includes an aligning torque, which depends on the local configuration of obstacles (Methods)[43], in the equations of motion of the active particles. The effect of this torque is to steer the active particles away from the surrounding passive ones and align their direction of motion to any effective boundary of a transient path[39,40]. Figure 4a shows how the presence of this aligning torque is already fundamental to reproduce the overall dependence of the experimental path revival lifetime $\tau_{\rho_p}$ on $\rho_p$, indicating that obstacle avoidance is indeed the mechanism that promotes active particles to follow previously formed transient paths. In fact, in the absence of the torque, the path revival lifetime increases with $\rho_p$ as one would intuitively expect due to the decrease of the particles' effective

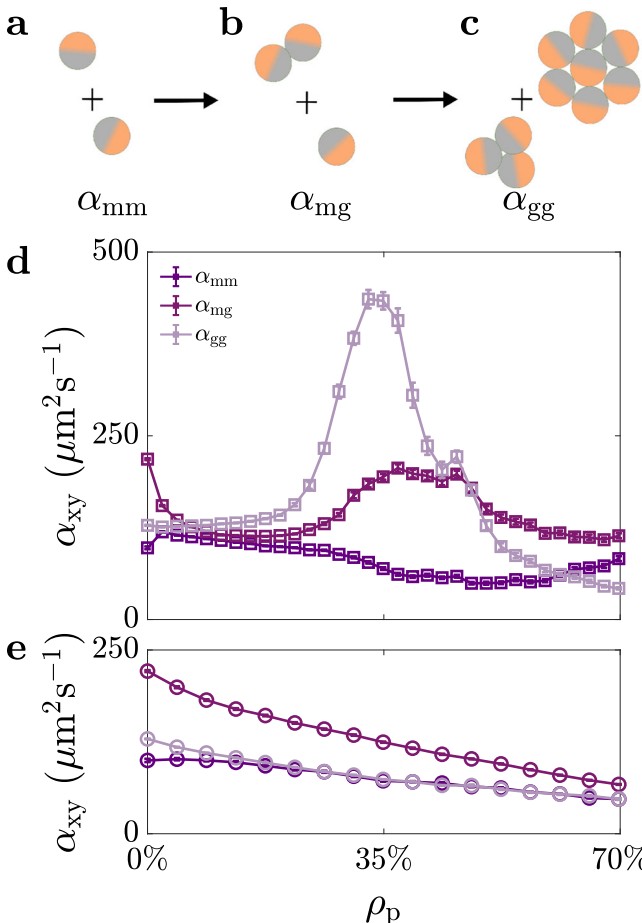

**Fig. 5 | Kinetics of group formation with shared environmental memory.**
**a**–**c** Schematics of the three main mechanisms for the kinetics of group formation described by the relevant rate coefficients $\alpha_{xy}$: **a** monomer–monomer ($\alpha_{mm}$), **b** monomer–group ($\alpha_{mg}$) and **c** group–group ($\alpha_{gg}$) aggregation. **d**, **e** Calculated monomer–monomer ($\alpha_{mm}$), monomer–group ($\alpha_{mg}$) and group–group ($\alpha_{gg}$) aggregation rate coefficients at a density of active particles $\rho_a = 1.1\%$ as a function of the density of passive particles $\rho_p$ (**d**) in the presence of an aligning torque and (**e**) in its absence. Simulations were averaged over 100 numerical experiments per value of $\rho_p$. Error bars represent one standard error around the average values. Source data are provided as a Source Data file.

velocity caused by collisions with the obstacles[41]. Fitting our experimental data to our model allows us to determine the strength of the torque to be $\Omega_0 = 72 \pm 16\, k_B T$. As shown in Fig. 4b, the presence of this torque is also critical to recovering the non-monotic dependence of the largest group size with $\rho_p$ as observed in our experimental data (Fig. 2b). Therefore, this torque and the resulting aligning interaction provide an enabling mechanism for the emergence of stigmergy via a shared environmental memory in the system of non-communicating active particles by allowing the spatial correlations in the environment to feed back on their motion. Figure 4b also confirms that the non-monotonic dynamics of group formation depend on the number of individuals (as already observed in Fig. 2b). At higher values of $\rho_a$, encounters become more probable so that groups can form and grow to larger sizes at lower values of $\rho_p$. As groups grow to larger sizes, there is a decrease in the density of passive particles needed to cage them and prevent them from merging into even larger groups. The combination of these two effects translates into a decreasing monotonic dependence of the peak position on $\rho_a$ (Fig. 4c), highlighting the relevance of environmental memory effects for group formation in sparse systems of clueless active particles. Figure 4d–f shows example

snapshots from the simulations, which confirm our qualitative observations in Fig. 2: at low values of $\rho_p$ (Fig. 4d), groups of a few units are formed; at intermediate values of $\rho_p$ (Fig. 4e), the landscape is dominated by a very few large groups that collect most of the active particles; finally, at large values of $\rho_p$ (Fig. 4f), a few relatively smaller groups of more homogeneous size appear to be caged within the crowded environment.

## Kinetic model of group formation

To further understand how the shared environmental memory affects encounter dynamics, we can define a kinetic model based on mean-field rate equations for the number density of monomers (free active particles, i.e., not part of a group) $c_1$ and for the number density of groups $c_g$ as

$$\dot{c}_1 = -\alpha_{mm} c_1^2 - \alpha_{mg} c_1 c_g, \tag{1}$$

$$\dot{c}_g = \frac{\alpha_{mm}}{2} c_1^2 - \frac{\alpha_{gg}}{2} c_g^2, \tag{2}$$

where $\alpha_{mm}$, $\alpha_{mg}$ and $\alpha_{gg}$ are the rate coefficients of monomer–monomer, monomer–group and group–group aggregation (Methods). Rate equations are indeed a powerful tool to understand the emergence of group dynamics and self-assembly in systems of multiple units, as much as in chemical kinetics[44] and colloidal science[12] as in swarm robotics[45,46]. In Eq. (1), monomers disappear due to the formation of a new group from two monomers (first term, Fig. 5a) or due to the growth of an existing group by addition of a new monomer (second term, Fig. 5b). Similarly, in Eq. (2), $c_g$ can change due to the formation of a new group from monomers (first term, Fig. 5a) or from the merging of two existing groups (second term, Fig. 5c). In all cases, we assume that the rate of encounters is proportional to the product of the number densities of the species involved (either monomers or groups) and that any dependence on the effective cross-sectional area of each species is accounted for by the effective rates of aggregation $\alpha_{mm}$, $\alpha_{mg}$ and $\alpha_{gg}$. Without a shared environmental memory, these rates should only depend on the effective diffusion coefficients of the species involved[47]. The larger the effective diffusion coefficient, the faster the rate of group formation and growth, leading to larger groups within the same time interval. Nonetheless, our experimental results suggest that the effective diffusion coefficients decrease with $\rho_p$ (Supplementary Fig. 2), so the augmented group formation for intermediate values of $\rho_p$ must result from the presence of spatial correlations in the environment that increase the chances for particles to meet, i.e., the reuse of transient paths highlighted in Fig. 1 and Supplementary Fig. 1.

By calculating the effective rate coefficients ($\alpha_{mm}$, $\alpha_{mg}$ and $\alpha_{gg}$) from the simulated data, we can assess the relative importance of monomer–monomer, monomer–group and group–group aggregation on the kinetics of group formation (Fig. 5d). Both $\alpha_{mg}$ and $\alpha_{gg}$ present a maximum for intermediate values of $\rho_p$ when group formation is enhanced, while $\alpha_{mm}$ is roughly constant in comparison. In the presence of passive particles, although monomer–monomer aggregation is key for the formation of the initial groups, the kinetics are dominated by groups catalysing their own growth through the addition of new monomers and merging with other existing groups. These aggregation events mediated by the presence of the shared environmental memory are hence behind the enhanced group formation observed at intermediate $\rho_p$. Indeed, when the aligning torque is switched off in simulation (Fig. 5e), the shared memory and stigmergy cannot develop (Fig. 4), resulting in aggregation rates that decay monotonically with increasing values of $\rho_p$, as one would expect when an increased number of obstacles hinders diffusion (Supplementary Fig. 2).

## Discussion

In summary, our results demonstrate how, in a decentralised system composed of clueless active units with no explicit signalling pathway or information-processing capability, a dynamic environment can create the conditions for the emergence of a shared environmental memory that can coordinate and shape the system's collective response. Hence, confinement by crowding becomes a condition sufficient for self-organisation to emerge and to activate the system's coordination capabilities (e.g., by naturally evolving to larger groups)[48]. In our experiments, the physical mechanism behind the emergence of these population dynamics is the aligning torque acting on the Janus particles due to the asymmetric demixing of the water−2,6-lutidine critical mixture caused by the presence of the non-fixed obstacles. Other physical mechanisms (e.g., electrostatic interactions, phoretic interactions, hydrodynamic coupling[39] or, even, sensory perception[32,48]) can be expected to lead to the emergence of a shared field memory in other systems, which could also be reproduced in a coarse-grained manner by introducing an effective aligning torque in their motion dynamics. Indeed, similar mechanisms of shared memory which are primarily promoted and reinforced by dynamic environmental factors could also contribute to shaping the collective dynamics of other decentralised systems where individuals can instead actively signal to each other[27], such as communities of micro-organisms[2,29], social insects[3,31] and robotic swarms[4,7]. The feedback from the environment could then lower the threshold for quorum formation in natural communities[2] and for reaching consensus in decision-making processes[26], e.g., by synergistically catalysing any pathway of explicit communication. Finally, we envisage that shared memories promoted by environmental dynamic features could become design factors to implement low-level rules to drive high-level self-organisation in artificial systems, including in the design of antimicrobial surfaces, of crowd management control tools, and of neuromorphic computers and artificial swarm intelligence[48].

## Methods

### Materials

Glass microscopy slides (Thermo Fisher) were purchased from VWR while glass coverslips were purchased from Thorlabs. The following chemicals were purchased and used as received: 2,6-lutidine (≥99%, Sigma-Aldrich), acetone (≥99.8%, Sigma-Aldrich), ethanol (≥99.8%, Fisher Scientific), sodium hydroxide (NaOH, Fisher Scientific). Deionised (DI) water (≥18 MΩ.cm) was collected from a Milli-Q purification system. Aqueous colloidal dispersions (5 % w/v) of silica (SiO$_2$) colloids for sample preparation (4.77 ± 0.20 µm in diameter for Janus particle fabrication and as passive colloids; 7.00 ± 0.15 µm in diameter as spacers) were purchased from Microparticles GmbH. Carbon rods of length 300 mm and diameter 6.15 mm for coating Janus particles by sputtering were purchased from Agar Scientific and cut to a length of 50 mm before use. Lens tissue for slide cleaning was purchased from Thorlabs. UV cure adhesive (Blufixx) and hydrophobic coating (RainX) for sample preparation were purchased from an online retailer (Amazon).

### Slide cleaning protocol

Before their use for sample preparation, glass slides and coverslips were cleaned by wiping them with acetone-soaked lens tissue. RainX (a commercial solution which renders glass surfaces more hydrophobic and aids in limiting particles sticking to the glass chamber) was then smeared on both with a cotton bud and gently dried with a nitrogen gun. After 2 min, excess RainX was removed by wiping with acetone-soaked lens tissue. Glass slides for the deposition of colloidal monolayers were instead cleaned by sonication for 10 min in a 2 M NaOH ethanolic solution followed by three cycles of 5 min sonication in DI water. To dry them, the slides were withdrawn from water in the presence of ethanol vapour (Marangoni drying) and, subsequently, blown dry with a nitrogen gun.

### Fabrication of Janus particles

The Janus particles used in our experiments were fabricated from SiO$_2$ colloids of diameter $d = 4.77 \pm 0.20$ µm, which were coated on one side with a thin layer ($\approx$60 nm) of carbon. We first deposited a monolayer of colloids on a clean glass slide. The monolayer was obtained by evaporating a 40 µL droplet containing a 2.5% w/v dispersion of the colloids in DI water. The particles were then coated with a 60 nm thick carbon layer using an automatic carbon coater (AGB7367A, Agar Scientific). The thickness of the carbon layer was confirmed by atomic force measurements (AFM). Post-coating sonication allowed us to dislodge the half-coated particles in DI water from the glass slide to use them for sample preparation.

### Sample preparation

Samples were prepared in the form of a quasi-two-dimensional glass chamber filled with a colloidal dispersion in a critical mixture of water−2,6-lutidine. Typical colloidal dispersions include Janus particles as well as passive SiO$_2$ particles and spacers. For example, to achieve a typical dispersion with densities of $\rho_a = 0.5\%$ and $\rho_p = 12.5\%$, we mixed stock dispersions of the three types of particles in DI water to achieve an aqueous dispersion containing 0.13% w/v of Janus particles, 5% w/v of passive particles and 0.08% w/v of spacers. This concentration of spacers was chosen to minimise their number in the field of view, whilst giving enough support to maintain the quasi-two-dimensional chamber's geometry. Samples with other densities ($\rho_a$ and $\rho_p$) were obtained by linearly scaling these concentrations of Janus particles and passive particles to obtain the right values of $\rho_a$ and $\rho_p$. Before their use, the colloidal mixtures were centrifuged at $1000 \times g$ for 3 min leaving a pellet; the supernatant was then removed and replaced with a 28.6% w/v water−2,6-lutidine solution. This process was repeated three times to remove residual DI water from the initial dispersion. Experiment-ready quasi-two-dimensional sample chambers containing a dispersion of colloids in a critical water−2,6-lutidine solution were prepared by sandwiching 10 µL of this final dispersion between a clean glass slide and a thin coverslip. The chamber was sealed by applying an ultraviolet-curable adhesive around the edges of the coverslip, which was then exposed to ultraviolet (UV) light for 30 s on each side. Before data acquisition, the sample was left to equilibrate over a 30-min period.

### Optical setup and microscopy

All the experiments were performed on an inverted microscope (Leica, DMI4000) equipped with a homemade flow thermostat to maintain the critical suspension at a fixed temperature ($T = 30$ ˚C) below the critical point ($T_c \approx 34.1$ ˚C). The sample's field of view was illuminated at once with a green continuous-wave laser ($\lambda = 532$ nm) at a power density of $2.5$ µW µm$^{-2}$ to simultaneously propel the Janus particles due to light absorption at the carbon cap[35]. Both Janus and passive particles were tracked by digital video microscopy[49] using the image projected by a microscope objective ($\times 20$, NA = 0.5) on a monochrome complementary metal-oxide-semiconductor (CMOS) camera (Thorlabs, DCC1545M) with an acquisition rate of 10 frames per second. The incoherent illumination for the tracking is provided by a white-light-emitting diode (Thorlabs, MWWHLP1) directly projected onto the sample. A long-pass dichroic mirror (Thorlabs, DMLP605) with a 605-nm cut-on wavelength was used to combine laser and white light before the sample, while laser light was removed from the detection path with a notch filter centred at 532 nm (Semrock, NF01-532U-25).

### Path revival function

To quantify the path reuse by the Janus particles, we define the path revival function $1 - C_{aa}(\tau)$, where $C_{aa}(\tau)$ is the cumulative probability that

a region crossed by an active particle will be crossed by another particle within a lag time $\tau$. To compute $1 - C_{aa}(\tau)$ we define a circular region of diameter $d$ around each active particle at a certain time $t$ and measure how many of those regions have been crossed by the centre of another active particle up to lag time $\tau$. If we consider the particles' velocities to be Poisson distributed when a path is chosen, then this function should follow an exponential distribution for persistent particles[41]

$$1 - C_{aa}(\tau) = \exp\left(-\tau/\tau_{\rho_p}\right), \qquad (3)$$

where $\tau_{\rho_p}$ is the effective path revival lifetime, which we fit from the data. For both experiments and simulations, we assume that the initial positions in the particles' trajectories are uncorrelated (i.e., in the experiments, we only consider trajectories of individual particles before groups form and, in the simulations, the short-range attractive interaction between particles is turned off to prevent group formation).

## Particle-based simulations

We consider a numerical model where $n_a$ active spheres and $n_p$ passive spheres of diameter $d$ move inside a two-dimensional box of side $L = 60d$ with periodic boundary conditions. Both $n_a$ and $n_p$ are fixed to match the experimental values of $\rho_a$ and $\rho_p$. As in the experiments, all particles, whether active or passive, have the same size $d$ and mass $m$.

To map the simulations to the experiments, we consider the same Péclet number defined as,

$$\text{Pe} = \frac{dv}{D_t}, \qquad (4)$$

where $d = 4.77\,\mu m$, $v = 1.9\,\mu m\,s^{-1}$, and $D_t = 0.0249\,\mu m^2\,s^{-1}$. Both velocity $v$ and diameter $d$ of the active particles were used to convert the reduced units in simulations to SI units. The translational diffusion coefficient $D_t$ was calculated as

$$D_t = \frac{k_B T}{\gamma'}, \qquad (5)$$

where $k_B$ is the Boltzmann constant, $T$ the absolute temperature, and $\gamma'$ is the corrected translational drag coefficient for colloids at distance $s$ from a surface[50], given by

$$\gamma' = \frac{\gamma}{1 - (9/16)(d/2s) + (1/8)(d/2s)^3}, \qquad (6)$$

with

$$\gamma = 3\pi\mu d, \qquad (7)$$

where $\mu$ is the fluid viscosity. We assume $2s = d$, $T = 303\,K$, and $\mu = 2.1 \times 10^{-3}\,Pa\,s$ for the water–2,6-lutidine mixture. Similarly, the rotational diffusion coefficient $D_r$ was calculated as

$$D_r = \frac{k_B T}{\beta'}, \qquad (8)$$

where $\beta'$ is the corrected rotational drag coefficient for colloids at distance $s$ from a surface[50], given by

$$\beta' = \frac{\beta}{1 - (1/8)(d/2s)^3}, \qquad (9)$$

with

$$\beta = \pi\mu d^3. \qquad (10)$$

The trajectories of both active and passive particles were obtained by integrating their equations of motion using a velocity Verlet scheme implemented in the large-scale atomic/molecular massively parallel simulator (LAMMPS)[51]. Specifically, the particles' translational motion and rotational motion around one single axis (perpendicular to the simulation plane) are respectively described by the following Langevin equations,

$$m\dot{v}_i(t) = -\nabla_{r_i} V_i - \frac{m}{\tau_\gamma}v_i(t) + \sqrt{\frac{2mk_B T}{\tau_\gamma}}\xi_t^i(t) + F_a\hat{u}_i(t) \qquad (11)$$

and

$$I\dot{\omega}_i(t) = \Omega_i - \frac{\alpha I}{\tau_\gamma}\omega_i(t) + \sqrt{\frac{2\alpha I k_B T}{\tau_\gamma}}\xi_r^i(t), \qquad (12)$$

where $v_i$ and $\omega_i$ are the translational and angular velocity for particle $i$, $\hat{u}_i = (\cos\theta_i, \sin\theta_i)$, $\omega_i = \dot\theta_i$, $F_a$ is the strength of the self-propulsion force for the active particles, $\tau_\gamma$ is the damping time, $I$ is the particles' inertia of rotation, $V_i$ is the potential due to the interaction with all surrounding particles, and $\Omega_i$ is an effective torque due to the interaction of particle $i$ with the surrounding passive particles. $\xi_t^i(t)$ and $\xi_r^i(t)$ are stochastic terms taken from a truncated random distribution of zero mean and unitary standard deviation[52]. Moreover, $\alpha$ is a model parameter that defines the relationship between the rotational ($D_r$) and translational ($D_t$) diffusion coefficients as

$$\frac{D_t}{D_r} = \alpha \frac{I}{m}. \qquad (13)$$

where $\alpha$ is adjusted to map the experimental relation between $D_t$ and $D_r$.

The motion of the passive particles is only governed by Eq. (11) (where we set $F_a = 0$) for computational efficiency as the rotational degree of freedom of the passive particles does not influence the numerical dynamics in the overdamped regime.

The interaction between particles is implemented with a Lennard-Jones potential given by

$$V_i = \sum_j V_{ij}(r_{ij}) = \sum_j 4\epsilon_{LJ}\left[\left(\frac{\sigma_{LJ}}{r_{ij}}\right)^{12} - \left(\frac{\sigma_{LJ}}{r_{ij}}\right)^6\right], \qquad (14)$$

where $r_{ij} = \|r_i - r_j\|$ is the distance between two particles, $\epsilon_{LJ}$ the depth of the potential well, and $\sigma_{LJ}$ the width of the potential (distance at which the potential is zero). For passive particles, with purely repulsive interactions, we consider a truncated Lennard-Jones potential where the cut-off is set at $r_{cut} = d = 2^{1/6}\sigma_{LJ}$ to remove the attractive part. For active particles, we consider an attractive interaction with a cut-off set at $r_{cut} = 5d$. The depth of the potential well $\epsilon_{LJ}$ is obtained from experimental data (Supplementary Fig. 4).

Finally, to describe the impact of the passive particles on the rotational degrees of freedom of the active particles, we introduce the effective torque $\Omega_i$ on particle $i$[43]

$$\Omega_i = -\Omega_0 d^2 \hat{v}_i \times \sum_{j=1}^{n_p} \nabla_r \frac{e^{-\kappa r_{ij}}}{r_{ij}}, \qquad (15)$$

where $\Omega_0$ sets the strength of the torque, $\hat{v}_i = v_i/\|v_i\|$ and $\times$ is the cross product. The negative sign indicates that active particles steer away from the surrounding passive ones; $\kappa = 0.25/d$ gives the screening number in agreement with the range of experimental values estimated in[43]. For numerical efficiency, we set a cut-off radius of four particle

diameters, where the value of the torque is much lower than the typical thermal noise. The torque used to map the experiments ($\Omega_0 = 72 \pm 16\,k_BT$) was computed and confirmed from two different experimental measurements (Figs. 4a,c). In our simulations, the interactions of the active particles with the passive obstacles are therefore dependent on both the potential $V_i$ and the torque $\Omega_i$, while the interaction between active particles only depends on $V_i$ as, working at low densities of active particles, they primarily interact when in groups.

## Rate equations

The relevant mechanisms for the dynamics are (1) the formation of new groups by combining two free active particles (monomers); (2) the growth of a group by the addition of a monomer; (3) the pairwise merging of groups; and (4) their fragmentation. In the experiments, we define groups as clusters of size larger than two as dimers are unstable in time. Here, for completeness, we consider all cases. We assume that groups only lose one active particle at a time (fragmentation).

We define $c_1$ and $c_i$ as the number densities of free active particles (monomers) and groups of size $i > 1$, respectively. The following rate equations then give the time evolution of $c_1$,

$$\dot{c}_1 = -\alpha_{mm}c_1^2 - \alpha_{mg}c_1\sum_{j>1}c_j + \sum_{j>1}f_jc_j + f_2c_2, \tag{16}$$

where the first term accounts for the formation of new groups, the second one for the growth of an existing group, the third for fragmentation, and the additional fourth term for the second free active particle released from the fragmentation of groups of size two. If the distance between groups is larger than the persistence length of the free active particles, the main mechanism of mass transport is diffusion and, in the absence of spatial correlations, the rates $\alpha_{mm}$ and $\alpha_{mg}$ should only depend on the size of the active particles and on their effective diffusion coefficients[47]. For simplicity, we also consider that $\alpha_{mg}$ does not depend on the group size.

Similarly, for groups of size two,

$$\dot{c}_2 = \frac{\alpha_{mm}}{2}c_1^2 - \alpha_{mg}c_1c_2 - \alpha_{gg}c_2\sum_{j>1}c_j - f_2c_2 + f_3c_3, \tag{17}$$

and for groups of size $k$,

$$\dot{c}_k = \alpha_{mg}c_1c_{k-1} - \alpha_{mg}c_1c_k + \frac{1}{2}\alpha_{gg}\sum_{i+j=k}c_ic_j - \alpha_{gg}c_k\sum_{j>1}c_j$$
$$+ f_{k+1}c_{k+1} - f_kc_k. \tag{18}$$

If we now define the number density of groups $c_g = \sum_{j>1}c_j$ and the total fragmentation rate $f = \sum_{j>1}f_jc_j$, we obtain,

$$\dot{c}_1 = -\alpha_{mm}c_1^2 - \alpha_{mg}c_1c_g + f + f_2c_2, \tag{19}$$

and,

$$\dot{c}_g = \frac{\alpha_{mm}}{2}c_1^2 - \frac{\alpha_{gg}}{2}c_g^2 - f_2c_2. \tag{20}$$

In the simulation, we observe that the total fragmentation rate is constant for a large range of $\rho_p$ up to the intermediate values where the largest groups are observed and then drops fast at higher values (Supplementary Fig. 5). Thus, aggregation rather than fragmentation is the leading factor in the non-monotonic dynamics of group formation observed in Fig. 2. If we neglect fragmentation, we obtain Eqs. (1) and (2), respectively.

## Data availability

All data supporting the findings of this study are available in the manuscript, the Supplementary Information and in the Source Data file. Source data are provided with this paper.

## Code availability

The code that supports the findings of this study is available from the corresponding authors on request.

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

## Acknowledgements

We are grateful to Samantha Rueber and Matthew Blunt for initial training on experimental techniques. C.S.D. and N.A.M.A. acknowledge financial support from the Portuguese Foundation for Science and Technology (FCT) under Contracts no. PTDC/FIS-MAC/5689/2020, EXPL/FIS-MAC/0406/2021, CEECIND/00586/2017, UIDB/00618/2020, and UIDP/00618/2020. G.V. (Giovanni) acknowledges funding from the Horizon Europe ERC Consolidator Grant MAPEI (Grant No. 101001267) and the Knut and Alice Wallenberg Foundation (Grant No. 2019.0079). G.V. (Giorgio) acknowledges sponsorship for this work by the US Office of Naval Research Global (Award No. N62909-18-1-2170). N.A.M.A. and G.V. (Giorgio) acknowledge support from the UCL MAPS Faculty Visiting Fellowship programme.

## Author contributions

Author contributions are defined based on the CRediT (Contributor Roles Taxonomy) and listed alphabetically. Conceptualisation: G.V. (Giovanni), N.A.M.A., G.V. (Giorgio). Data curation: C.S.D., M.T. Formal analysis: C.S.D., M.T., N.A.M.A., G.V. (Giorgio). Funding acquisition: G.V. (Giorgio). Investigation: C.S.D., M.T., N.A.M.A. Methodology: C.S.D., M.T., G.V. (Giovanni), N.A.M.A., G.V. (Giorgio). Project administration: G.V. (Giorgio). Resources: N.A.M.A., G.V. (Giorgio). Software: C.S.D. Supervision: C.S.D., N.A.M.A., G.V. (Giorgio). Validation: C.S.D., M.T., G.V. (Giorgio). Visualisation: C.S.D., M.T., G.V. (Giorgio). Writing—original draft: C.S.D., G.V. (Giovanni), NAMA, GV (Giorgio). Writing—review and editing: C.S.D., M.T., G.V. (Giovanni), NAMA, G.V. (Giorgio).

## Funding

## Competing interests

The authors declare no competing interests.
