## [Peer Review File · Nature Communications]

REVIEWER COMMENTS

Reviewer #1 (Remarks to the Author):

In this paper the authors have found an interesting stigmergy of active particles in colloids, where group formation is observed mediated by environmental memory. Not only the phenomenon itself but their interpretation of the mechanism is surely new, but the discussions shown in this paper are not satisfactory in the physical point of view which is explained below.

1) I admit that the torque is especially important for the group formation and reuse the path for following particles. But the physical mechanism is still unclear. Why the torque contributes to align the moving direction for active particles, and how this contributes the obstacle avoidance? Since introduction of rotation of an active particle makes the motion complex and may result in increasing collisions between passive particles in some cases.

2) The reuse of path may be interpreted in a different way, e.g., by using the concept of “acoustic streaming”. If an active particle digs into the sea of passive particles, then the background water is nonlinearly induced to form a channel flow afterwards. Then it becomes possible that the following particles use the channel. In the authors model, the motion of the background water is not considered explicitly. Thus the role of water should also be discussed for their analysis of the phenomenon.

3) What is the definition of “stable cluster” used in this paper? Active particles lose the energy by the collisions, but they keep moving due to the Brownian diffusion. Then the particles in a cluster are sometime detached each other in the course of time. For example, in Fig.3e, we see a large cluster which is not perfectly caged by passive particles. Then the peripheral particles may sometimes be detached from the cluster due to fluctuations.

4) In Fig.S3b, the curve is not monotonously decreasing with ρ_p . Especially at 25% and 50%, we see the rise of the $\tau_{\rho p}$ which contradicts the interpretation given in this paper. We cannot omit these cases in Fig.S3a.

5) In “Particle-based simulations”, the passive particle is govern only by eq.(11) and neglecting the rotation, which is different from the case of active particles. I’m afraid that rotation energy may unphysically be treated in the model since there is collision between active and passive particles.

Also in eq.(15), is the sum done over the passive particles only? There is friction between active particles, which can also produce torque.

Typos

#The last line in page 6, “larger groups lowers” may be “larger groups”.

#In eq.(16), the lowerscript in the sum may be $j > 2$ instead of $j > 1$, since the $j=2$ case is identical to the last term in (16).

Reviewer #2 (Remarks to the Author):

In the paper “Environmental Memory Boosts Group Formation of Clueless Individuals,” the authors study the aggregation dynamics of active particles in environments crowded by passive particles. The authors perform experiments with real particles and derive a particle-based simulation model that is then further used to compute certain system-level characteristics and a rate equation model.

The swarm intelligence aspect of the work is very interesting and highlights how principles of self-organization emerge and can be exploited at microscopic scales. However, I am lacking the technical knowledge in material science to evaluate the correctness, novelty and impact in that field.

The manuscript is overall well written and explains well the work conducted and results obtained. Yet, there exist issues that I believe may require some more work.

The emergent effect is classified as stigmergy by the authors. This may be argued based on common definitions. Usually stigmergy is defined as “communication by means of the environment” and is asynchronous. Given that here the initial particle is basically blazing a trail that significantly simplifies motion already for the 2nd particle following it (unlike -say- trail formation by humans on low grass), this may be seen rather as construction of a trail/street. However, this is a potentially tedious definition discussion. I could accept the classification of the effect as stigmergy but maybe the authors can argue or rethink.

On page 5, the authors state that “While, for a given ρ_a , the average size of these groups seems roughly constant (Fig. 2d)”, however, Figure 2d shows that the average size of the groups varies both in ρ_a and ρ_p . This seems a question of dynamics/differences living on different scales? Maybe this could be made more explicit in the paper.

The authors mention they used an aligning torque as assumption and implementation in the simulation to account for the emergence of path reuse. Yet, it is not clear what motivates the exclusive focus on torque, except for a post-hoc rationalization through experiments. This might be the simplest hypothesis or the only possible hypothesis? It would be beneficial if the authors could expand upon their justifications of modeling the emergence as torque.

The effects shown in Figs. 2b, 3b and the rate equation model seem to align well with other results from previous work, such as Hamann & Reina, Scalability in Computing and Robotics, 2021 also the rate equation model can be related to common approaches in swarm robotics. The authors may consider to draw the obvious connections also to related domains here.

Minor comment:

- It is not clear to me if all Janus particles are light activated at the same time or sequentially? A clarification might be useful. I understand that due to possibly very different timescales of activation and motion the sequential activation may not introduce any problems.

Reviewer #3 (Remarks to the Author):

This is a well written paper on the interesting topic of stigmergic memory in a collective system of clueless (processing-less) individuals.

I think the paper is quite mature in its current form and I have only a few minor comments.

My main comment is regarding the "path revival lifetime" that is used in pages 5 and 6. The path revival function is explained already in the main text, but the "path revival lifetime" is not defined before the Methods section, which hinders the understanding of the discussions that follows (and the related figures). I think a clear description is necessary in the main text - maybe you can use the equation 3 and a short explanation on what it is.

Most of the links to figures S1, S2, S3 bring the reader to figures 1, 2, and 3 instead.

Fig 3: What is survival time mentioned in the caption?

P6: is there a repulsion between the active and passive particles that lead to the aligning torque, or what is the cause of the torque?

Manuscript NCOMMS-23-25939
Authors' Response to Reviewers' Comments

We are grateful to all Reviewers for carefully reading our manuscript and for their positive feedback praising the interest and quality of our work. The points raised by the Reviewers have helped us clarify important aspects of our work and to stress its novelty even further. We have thoroughly addressed all these points in the revised version of the manuscript. Our point-by-point response to the Reviewers' comments follows below. All changes are highlighted in red in the revised version of the paper.

Please, note that, due to journal formatting requirements, all figures now follow the references and end notes in the new version of the manuscript.

Reviewer #1 (Remarks to the Author):

In this paper the authors have found an interesting stigmergy of active particles in colloids, where group formation is observed mediated by environmental memory. Not only the phenomenon itself but their interpretation of the mechanism is surely new, but the discussions shown in this paper are not satisfactory in the physical point of view which is explained below.

We thank the Reviewer for recognising the originality of our results and their interpretation. We have improved the discussion of the physical mechanism behind our results addressing the issues identified by the Reviewer. Beyond expanding the discussion, we performed further data analysis and simulations to support it. We detail our changes below.

1) I admit that the torque is especially important for the group formation and reuse the path for following particles. But the physical mechanism is still unclear. Why the torque contributes to align the moving direction for active particles, and how this contributes the obstacle avoidance? Since introduction of rotation of an active particle makes the motion complex and may result in increasing collisions between passive particles in some cases.

We thank the Reviewer for highlighting the importance of the torque in the group formation and the need to clarify its contribution. Below, we detail the physical mechanism we identified to be responsible for the torque on the active particles near the non-fixed obstacles and how this leads to obstacle avoidance and alignment to existing transient paths.

Our Janus particles are constituted by two hemispheres with different physicochemical properties: a more hydrophilic silica side and a more hydrophobic carbon side. As can be seen in Fig. R1, they self-propel with the hydrophobic carbon cap at the front (darker half of the Janus particle) in a critical mixture of water and 2,6-lutidine, with water being the major component (0.286 mass fraction of lutidine). This mixture presents a lower critical temperature around 307 K above which a phase separation occurs between a water-rich phase and a lutidine-rich phase. When illuminated with enough intensity, light absorption at the particle's carbon cap induces a local demixing of the critical mixture in its surrounding, leading to the formation of a lutidine-rich droplet around the more hydrophobic carbon cap and of a smaller water-rich droplet around the more hydrophilic silica side. The formation of these droplets breaks the symmetry around the particle making it self-propel [see, e.g., Buttinoni *et al.*, J. Phys.: Condens. Matter 24, 284129 (2012); Gomez Solano *et al.*, Sci. Rep. 7, 14891 (2017)].

Figure R1. This time sequence shows how the motion of a self-propelled Janus particle experiences a torque that aligns it towards openings in the background of Brownian colloids. The effect of this torque on the particle's trajectory is evident when the colloid turns clockwise towards a void in the structure formed by the passive particles (between the two frames at $t = 24$ s and $t = 34$ s) and then counter-clockwise towards an open path (between the two frames at $t = 47$ s and $t = 58$ s). In each image, 24-s-long trajectories are shown for both active (red) and passive (blue) particles; t represents the time of each frame and τ the time along each trajectory (colour-codes, as shown in the colour bars on the right). Scale bar: 10 μm .

In the absence of obstacles, these droplets are mostly symmetric around the axis defined by the direction of the particle's motion and the particle's reorientation in a viscous medium is

mainly driven by its intrinsic rotational diffusion dynamics [see, e.g., Buttinoni *et al.*, J. Phys.: Condens. Matter 24, 284129 (2012); Gomez Solano *et al.*, Sci. Rep. 7, 14891 (2017)]. Asymmetric demixing around the direction of the particle's motion has already been demonstrated to induce deterministic aligning torques on similar carbon-coated Janus particles moving in light gradients. These torques lead to a large reduction of the randomness of the reorientational dynamics caused by rotational diffusion and contribute to strongly align the particle with the light gradient [see, e.g., Gomez Solano *et al.*, Sci. Rep. 7, 14891 (2017)].

In our case, it is the presence of obstacles (rather than of a light gradient) that leads to demixing being no longer symmetric around the direction of the particle's motion (because it is blocked on the obstacle side). This asymmetry induces a deterministic torque that reorients the particle towards the more demixed side (similar to what is observed in light gradients), that is, away from the obstacles (i.e., avoiding collisions) when moving with the carbon cap at the front. This reorientation (and, hence, the presence of the torque) depends on the local configuration of obstacles and continues until the particle is aligned in a direction where the demixing is symmetric again and the torque vanishes.

This can be clearly seen in Fig. R1. At $t = 24$ s, a Janus particle approaches a group of relatively packed obstacles. On reaching one of them, it turns clockwise ($t = 34$ s) towards a void in the structure formed by the passive colloids, instead of keeping on moving in the same direction of approach. Interestingly, in this case, although the particle could have turned right towards an open path (offering less resistance to its motion in principle), the clockwise reorientation towards the void confirms our hypothesis about the torque on the active particles being induced by the asymmetric demixing due to the presence of the obstacle on one side of the particle based on its direction of approach (in this case with the obstacle on its left side). At $t = 34$ s, the Janus particle approaches two obstacles via their middle (an approximately symmetric approach), thus pushing them out of the way and continuing its journey towards a previously opened path ($t = 47$ s). Nonetheless, the particle does not align with this path straightaway. It is not until later, when it encounters a new block of relatively packed obstacles, that a new reorientation event turns it counter-clockwise with the obstacle on its right side, thus aligning the particle's motion to the open path ($t = 58$ s) and allowing its reuse.

Figure R2. Distribution of deflection angles for Janus particles near obstacles. The histogram shows the probability $P(\delta)$ for the Janus colloids to be deflected at an angle δ by their interaction with the obstacles (inset) when approaching within a narrow cone of acceptance ($\pm\pi/8$ excluding near head-on approaches as they contribute symmetrically around $\delta = 0$); δ is the angle between the active particle's direction of approach to the obstacle (obtained at a distance between the two particles' surfaces equal to one diameter d) and its direction of motion after having passed the obstacle (after traveling a $2d$ distance). Positive values indicate deflection away from the obstacles (as shown in the inset), while negative angles indicate deflection towards the obstacles. To primarily focus on interactions with single obstacles, we computed this distribution at low density of passive particles ($\rho_p = 12.5\%$). The

solid line is a Gaussian fit around the position of the peak (centred at $\delta \approx 0.23$ rad, vertical dashed line).

To quantify the previous observation, we have calculated the distributions of angles δ by which the Janus colloids are deflected by their interaction with the obstacles (Fig. R2). This value is calculated as the angle formed between the direction of approach to the obstacle and the direction of motion of the particle after having passed it. Positive and negative values indicate that the particles are deflected initially away from or towards the obstacles, respectively. When steering away from the obstacles (positive δ), we can then expect the particles to almost immediately align to open areas in the background of passive particles, while deflections towards the obstacles (negative δ) would initially increase chances of collisions with the obstacle until forming a new path or aligning to an existing open area. The distribution of angles in Fig. R2 is peaked at positive angles, thus quantitatively confirming the stronger tendency for active particles to avoid obstacles and align to open paths already on approach.

To address these points, we have introduced the following changes to the manuscript:

- On page 3, we added: "...light absorption at the carbon cap simultaneously propels the Janus particles in the field of view with the more hydrophobic carbon-coated side at the front...".
- On pages 5-6, we included a detailed discussion of the physical mechanism behind the torque: "To shed light on the physical mechanism behind the reuse of paths in Fig. 1 and Supplementary Fig. 1, Figs. 3a-b show a close-up of the motion of a Janus particle in the background of passive particles. At $t = 24$ s (Fig. 3a), the particle approaches a block of relatively packed obstacles. On approaching one of them, it turns clockwise towards a void in the structure formed by the passive colloids instead of pushing ahead. At $t = 34$ s, the Janus particle approaches two obstacles head-on via their middle, thus pushing them out of the way and continuing its journey towards a pre-existing path (Fig. 3b). Interestingly, the particle does not align with this path until it encounters a new block of relatively packed obstacles ($t = 47$ s). At this point, a new reorientation event turns it counter-clockwise, thus aligning the particle's motion to the open path ($t = 58$ s) and allowing its reuse. To interpret these reorientation events, we need to consider how the presence of the obstacles affects the particle's self-propulsion mechanism. In their absence, light absorption at the particle's carbon cap induces the local demixing of the water–2,6-lutidine critical mixture, leading to the formation of a lutidine-rich droplet around the more hydrophobic carbon side and of a smaller water-rich droplet around the more hydrophilic silica side [35, 42]. The formation of these droplets, which propels the particle [35, 42], is mostly symmetric around the axis defined by the direction of the particle's motion, and the particle's reorientation in a viscous medium is driven by its intrinsic rotational diffusion dynamics [35, 42]. If obstacles are present on one side, the local demixing is no longer symmetric around the direction of the particle's motion. This asymmetry induces a deterministic aligning torque that reorients the particle towards the more demixed side, i.e., eventually away from the obstacles [42]. As can be seen in Figs. 3a-b, this reorientation (and, hence, the presence of the torque) depends on the local configuration of obstacles and continues until the particle is aligned in a direction where the demixing around the caps is symmetric again and the aligning torque vanishes (for example, along an open path). To quantify the previous observation, we have calculated the angle δ at which the Janus colloids are deflected by their interaction with the obstacles (Fig. 3c). Positive and negative δ values indicate that the particles are deflected initially away from or towards the obstacles, respectively. When steering away from the obstacles (positive δ), we can then expect the particles to immediately align to openings in the background of passive particles, while deflections towards the obstacles (negative δ) would initially increase chances of collisions with the obstacles until the particle digs a new path or aligns to an existing one. The distribution of angles

in Fig. 3c is peaked at positive δ values (as highlighted by the Gaussian fit centred at $\delta \approx 0.23$ rad), thus quantitatively confirming the stronger tendency for active particles to avoid obstacles and align to open paths already on approach.”

- On page 6, we rephrase the introduction of the model as: “... we can therefore consider a simple particle-based model that includes an aligning torque ... which depends on the local configuration of obstacles (Methods).”
- We added a new figure to include the data in Figs. R1 and R2 (Fig. 3 in the new version of the manuscript).
- We added Gomez-Solano *et al.*, Sci. Rep. 7, 14891 (2017) as new reference [42].

2) The reuse of path may be interpreted in a different way, e.g., by using the concept of “acoustic streaming”. If an active particle digs into the sea of passive particles, then the background water is nonlinearly induced to form a channel flow afterwards. Then it becomes possible that the following particles use the channel. In the authors model, the motion of the background water is not considered explicitly. Thus the role of water should also be discussed for their analysis of the phenomenon.

We agree with the Reviewer that, in principle, similar effects as those reported here could be observed independently of the microscopic physical mechanism that leads to the alignment of the particles and to the formation of transient paths. For example, alignment could also be caused by, e.g., electrostatic interactions, phoretic interactions, hydrodynamic coupling or, even, sensory perception in more advanced systems. In general, a torque could be introduced to model these effects. However, we have a more physical explanation for the mechanism at work: as explained in the previous point, the torque due to the asymmetric demixing around the Janus particles caused by the presence of the obstacles is the proper microscopic mechanism leading to alignment in our system.

Our experiments are at very low Reynolds numbers ($Re \approx 10^{-5} \ll 1$) and, therefore, we are in the Stokes regime. To the best of our knowledge, acoustic streaming is typically observed at significantly higher Reynolds numbers ($Re > 1$) and hence is not expected here. In the Stokes regime, fluid displacement caused by a particle would typically intervene at much shorter timescales (i.e., ms at most [see, e.g., Huang, R. et al., Nat. Phys. 7, 576–580 (2011) and Pesce, G. et al., Phys. Rev. E 90, 042309 (2014)]) than the dynamics observed in our experiments (in the order of seconds to tens of seconds, i.e., comparable with the particle’s rotational dynamics); therefore, its influence is usually negligible on such time scales. Indeed, in our experiments, we do not observe evidence of any flow patterns, even induced locally in the channels, of strength such to affect the overall particle’s dynamics. For example, if these flows were present, they would also induce the passive obstacles to drift in their direction, which we have never observed. We only observe the passive particles to move directionally when physically pushed by a Janus particle (see, e.g., the time sequences in Fig. 1a-d, Supplementary Fig. 1 and Fig. R1).

To clarify this important aspect, we have changed the following in the revised manuscript:

- On page 3, we added: “Our experiments are in the Stokes regime (Reynolds numbers, $Re \approx 10^{-5} \ll 1$) and inertial effects, including those of the fluid [36, 37], can be safely neglected.”
- Starting on page 3, we also added: “Although the presence of voids in the background of passive particles can simplify this task at times, their overall motility reduces for increasing values of ρ_p [...] Unless pushed by a Janus particle, the motion dynamics of the passive particles remain diffusive at the edges of the transient paths (Fig. 1a and Supplementary Fig. 1a).”
- In the conclusion on page 9, we added the following discussion: “In our experiments, the physical mechanism behind the emergence of these population dynamics is the aligning torque acting on the Janus particles due to the asymmetric demixing of the water–2,6-lutidine critical mixture caused by the presence of the non-fixed obstacles.

Other physical mechanisms (e.g., electrostatic interactions, phoretic interactions, hydrodynamic coupling [39] or, even, sensory perception [32, 49]) can be expected to lead to the emergence of a shared field memory in other systems, which could also be reproduced in a coarse-grained manner by introducing an effective aligning torque in their motion dynamics.”

- We included two additional references ([36-37] in the new version of the manuscript).

3) *What is the definition of “stable cluster” used in this paper? Active particles lose the energy by the collisions, but they keep moving due to the Brownian diffusion. Then the particles in a cluster are sometime detached each other in the course of time. For example, in Fig.3e, we see a large cluster which is not perfectly caged by passive particles. Then the peripheral particles may sometimes be detached from the cluster due to fluctuations.*

We apologise for our misleading use of the word “stable”, which we have now removed. We defined a cluster as any group of three or more particles separated by at most 10% of their diameter from another particle and surviving for at least one frame. This definition is consistent with similar definitions used in the literature to define clusters of active colloids [see, e.g., current reference 12]. We agree with the Reviewer that a cluster can gain or lose particles in time (as is often the case for aggregations of active colloids). Supplementary Figs. 4 and 5 show that aggregation is the leading dynamic in our system as a short-range attraction of $150k_B T$ between particles in a cluster overpowers translational Brownian motion ($\sim k_B T$). Moreover, the characteristic time scale for the rotational diffusion of our active colloids is relatively long: $\tau_r = 1/D_r \approx 196$ s, with D_r the particle’s rotational diffusion coefficient. This, together with the need of overcoming the short-range attraction, also reduces occurrences where particles can leave a cluster when pointing away from it due to their rotational Brownian motion.

To remove potential sources of confusion, we apported the following changes:

- We removed the word “stable” when discussing clusters.
- We added the following definition on page 3 when we first introduce the concept of groups: “**here defined as the formation of a cluster of at least three particles separated by at most $0.1d$ [note: d is the particle’s diameter] from another particle and surviving for at least one frame.**”

4) *In Fig.S3b, the curve is not monotonously decreasing with ρ_p . Especially at 25% and 50%, we see the rise of the τ_{ρ_p} which contradicts the interpretation given in this paper. We cannot omit these cases in Fig.S3a.*

We have now added the additional curves in the figure, which were originally omitted to improve its readability. We have also rephrased the discussion of Supplementary Fig. 3 on page 5 to account for these instances; it now reads: “Supplementary Figure 3a shows how, in our experiments, **after initial comparable trends at lower values of ρ_p ($\rho_p \leq 25\%$), the decay of the path revival function becomes faster starting from intermediate values of ρ_p (quantified by an approximately factor-two reduction of the path revival lifetime τ_{ρ_p} in Supplementary Fig. 3b) [...]** This **change** of the path revival lifetime [...]. **Eventually, further increasing ρ_p [...]**” We have also changed the caption of Supplementary Fig. 3 to reflect these changes.

5) *In “Particle-based simulations”, the passive particle is govern only by eq.(11) and neglecting the rotation, which is different from the case of active particles. I’m afraid that rotation energy may unphysically be treated in the model since there is collision between active and passive particles. Also in eq.(15), is the sum done over the passive particles only? There is friction between active particles, which can also produce torque.*

We thank the Reviewer for highlighting the need to clarify our assumptions regarding the rotational degrees of freedom of the passive particles. Both active and passive particles move in a viscous medium (Stokes regime, very low Reynolds numbers, $Re \approx 10^{-5} \ll 1$), where inertial effects can be safely neglected assuming overdamped motion for the particles (for the time and length scales considered in the manuscript). Any energy transfer to the rotational degrees of freedom of a passive particle will be dissipated through the interaction with the medium. Excluding the rotation of the passive particles in the equation of motion (whether caused by rotational Brownian motion or by a torque induced by nearby active/passive particles) will not impact on the dynamics in a qualitative or quantitative way.

To confirm this point, we performed new numerical simulations, where we explicitly considered the rotational Brownian motion of the passive particles and a torque acting on them. Fig. R3 shows results obtained from three simulations. In the first set of simulations (top of Fig. R3), the passive particles are subject to both rotational Brownian motion and the effect of the torque, which is calculated as in Eq. 15 over all nearby particles (both active and passive in this case); in the second set (middle of Fig. R3), the torque is switched off, but the rotational Brownian motion is maintained; finally, in the third set (bottom of Fig. R3), both effects are switched off as in the simulations reported in the paper. To exclusively focus on the effect of the torque and the rotational Brownian motion, we used the same time series for the noise terms in all three cases. As can be seen in the figure, the trajectories of active and passive particles are identical in the three cases, showing that despite the changes in the rotational degrees of freedom (as shown by the grey gradient representing the particles' orientation), there is no impact on the translational motion. This confirms that the rotational degrees of freedom of the passive particles can be decoupled, which significantly reduces the computational effort while still enabling to numerically access the relevant timescales and regions of interest in the parameter space of our experimental system.

Figure R3. 125-s simulations (only last 25 s shown for clarity) of active particles moving in a background of passive particles for the cases where, beyond translational Brownian motion

and a Lennard-Jones interaction, the equation of motion of the passive particles also include: (top) both rotational diffusion and an aligning torque due to nearby passive and active particles; (middle) rotational diffusion but no torque; and (bottom) neither of the two effects as in our original simulations. The grey scale represents the orientation of the passive particles in radians. Scale bar: 20 μm .

Finally, as the Reviewer correctly points out, Eq. 15 only considers the torque to depend on passive particles, but at no detriment for our analysis and simulations. The reason for this is threefold:

- first, the mechanism that we have identified to be responsible for the torque near passive particles (see our reply to point 1) does not apply near other Janus particles as heating at multiple carbon caps would not hamper local demixing but rather enhance it;
- second, the complex interaction among multiple active particles is already accounted for by the Lennard-Jones potential which is calibrated on our experiments (Supplementary Fig. 4). This has already been used to successfully explain the formation of cluster of active particles in homogenous environments at much higher densities than in our experiments. For example, see Buttinoni *et al.*, Phys. Rev. Lett. 110, 238301 (2013) for carbon-coated Janus particles in a water–2,6-lutidine critical mixture and current reference [38] for a more general study (i.e., independent of the specific propulsion mechanism);
- finally, we work at very low densities of active particles (1.6% at most in experiments), so active–active interactions are extremely rare before the formation of a group – thus the likelihood of these affecting the motion dynamics of individual particles (i.e., before the formation of cluster as considered in this work) is low and can be neglected.

To clarify these points, we have made the following changes to the manuscript:

- In the methods, on page 13, we specify: “The motion of the passive particles is only governed by Eq. 11 (where we set $F_a = 0$) for computational efficiency as the rotational degree of freedom of the passive particles does not influence the numerical dynamics in the overdamped regime.”
- In the methods, on page 14, we added: “In our simulations, the interactions of the active particles with the passive obstacles are therefore dependent on both the potential V_i and the torque Ω_i , while the interaction between active particles only depends on V_i as, working at low densities of active particles, they primarily interact when in groups.”

Typos

#The last line in page 6, “larger groups lowers” may be “larger groups”.

#In eq.(16), the lowerscript in the sum may be $j > 2$ instead of $j > 1$, since the $j=2$ case is identical to the last term in (16).

We thank the Reviewer for the suggestions. We have rephrased the two sentences to make them clearer. Specifically,

- We rephrased the last line on page 6 (now on page 7) of the previous manuscript to: “As groups grow to larger sizes, there is a decrease of the density of passive particles needed to cage them and to prevent them from merging into even larger groups.”
- The lower script in Eq. 16 is correct as $j > 2$ is an additional term to account for the fact that the fragmentation of a group of size two leads to the formation of two monomers, without this term mass would not be conserved. To remove potential sources of confusion, we have rephrased the definition after the equation from “... the fourth term [accounts] for the additional free active particle obtained from the fragmentation of

groups of size two” to “the additional fourth term [accounts] for the second free active particle released from the fragmentation of groups of size two.”

Reviewer #2 (Remarks to the Author):

In the paper “Environmental Memory Boosts Group Formation of Clueless Individuals,” the authors study the aggregation dynamics of active particles in environments crowded by passive particles. The authors perform experiments with real particles and derive a particle-based simulation model that is then further used to compute certain system-level characteristics and a rate equation model.

The swarm intelligence aspect of the work is very interesting and highlights how principles of self-organization emerge and can be exploited at microscopic scales. However, I am lacking the technical knowledge in material science to evaluate the correctness, novelty and impact in that field.

The manuscript is overall well written and explains well the work conducted and results obtained. Yet, there exist issues that I believe may require some more work.

We thank the Reviewer for recognizing the appeal of our work for swarm intelligence and self-organization as well as the quality of our write-up. We have addressed the remaining issues below.

The emergent effect is classified as stigmergy by the authors. This may be argued based on common definitions. Usually stigmergy is defined as “communication by means of the environment” and is asynchronous. Given that here the initial particle is basically blazing a trail that significantly simplifies motion already for the 2nd particle following it (unlike -say- trail formation by humans on low grass), this may be seen rather as construction of a trail/street. However, this is a potentially tedious definition discussion. I could accept the classification of the effect as stigmergy but maybe the authors can argue or rethink.

As the Reviewer pointed out, common definitions of stigmergy explain it as a mechanism of indirect coordination mediated by communication through the environment [see, e.g., current reference 27]. Stigmergy can be categorised as *sematectonic* or *marker-based* depending on whether the communication that triggers the population behaviour is respectively channelled by modifications of a physical environment or by a signalling mechanism (e.g., the pheromones deposited by social insects) [see, e.g., current reference 27]. In our case, as described by the Reviewer, the trails are initially formed by the passage of the first active particles, which move away the passive (non-fixed) obstacles. As the passive particles move due to thermal fluctuations or by being pushed by other active particles, these trails eventually fade (see, for example, differences between the first and last frame in Fig. R1). Their persistence in time is thus intrinsically connected with the ability of the active particles to reuse them frequently. Thus, our classification fits the most general use in the literature of the concept of *sematectonic stigmergy* [see, e.g., current reference 27], where modifications introduced by an agent in their physical environment (here, the formation of a path) act as asynchronous cues directing the next steps of other agents (here, changing the direction of motion of other active particles) in a way that provides a direct contribution to the task (here, the carving, reuse and stabilization of the paths). Following the Reviewer’s advice, we therefore prefer to argue our classification in the text, as we believe our work is of interest to the communities studying decentralized architectures and stigmergy in natural and robotic systems. Maintaining this classification will help cut language barriers across communities and reach the relevant audience for our work.

To raise awareness towards the point raised by the Reviewer, we now provide an explanation of why we classify our observations as stigmergy in the new version of the manuscript. In particular:

- We included the distinction between sematectonic and marker-based stigmergy in the introduction on page 2: “... **stigmergy is a form of indirect communication between**

individuals by means of the environment, either mediated by physical modifications (sematectonic stigmergy) or by a signalling mechanism via deposition of markers (marker-based stigmergy) which shape a shared environmental memory [27].”

- On page 4, we introduced the explanation: “A form of stigmergy (consistent with the definition of sematectonic stigmergy [27]) between the active particles is then established ... As in other cases of sematectonic stigmergy [27], therefore, the modifications introduced by the active particles in their physical environment (here, the formation of a path in the background of passive colloids) act as asynchronous cues directing the next steps of other particles (here, changing their direction of motion) in a way that provides a direct contribution to the task (here, the carving, reuse and stabilization of the paths) and facilitates the emergence of population-level coordination (here, group formation).”

On page 5, the authors state that “While, for a given ρ_a , the average size of these groups seems roughly constant (Fig. 2d)”, however, Figure 2d shows that the average size of the groups varies both in ρ_a and ρ_p . This seems a question of dynamics/differences living on different scales? Maybe this could be made more explicit in the paper.

We apologise for the confusion. We did observe trends in Fig. 2d that depend both on ρ_a and ρ_p . Initially, for lower values of ρ_p , we observe that groups tend to increase in average size until a peak is reached around intermediate values of ρ_p . The position and width of this peak depend on ρ_a : the peak is higher, broader, and shifted towards lower values of ρ_p , the larger ρ_a . This is indeed connected to differences in how the dynamics evolve with ρ_a and ρ_p as mentioned by the Reviewer. Indeed, as chances for encounter increase with the number of available active particles, individuals become more efficient at creating and reusing correlations in their environment through the transient paths, thus lowering the density threshold needed for the passive phase to promote group formation with increasing ρ_a . The behaviour past the peak (i.e., for higher values of ρ_p) was already discussed in detail on page 5 of the manuscript.

Following the Reviewer’s suggestion, we have now amended and expanded the discussion of Fig. 2d in the text. In particular,

- on pages 4-5, we added: “The average size of these groups also shows a tendency to increase for a given ρ_a until a peak is reached around intermediate values of ρ_p (Fig. 2d). Both the exact position of this peak and its width depend on ρ_a : the peak is higher, broader, and shifted towards lower values of ρ_p , the higher ρ_a (Fig. 2d). This is also reflected by the trend observed for C_{max} (Fig. 2b). Indeed, as chances for encounter increase with the number of available active particles, individuals become more efficient at creating and reusing correlations in their environment through the transient paths, thus lowering the density threshold needed for the passive phase to promote group formation with increasing ρ_a . At the peak, a larger cluster tend to emerge (Fig. 2b) that can contain up to $\approx 67\%$ of the particles in a group (n_{pg}) due to the shared environmental memory from the path reuse highlighted in Fig. 1 and Supplementary Fig. 1. This effect is more prominent the higher the value of ρ_a .”
- on page 5, we changed “...translates now into smaller groups of more homogeneous sizes and more localised in space, which ... are roughly independent of the initial value of ρ_a ...” to “...translates now into smaller groups of more homogeneous sizes and more localised in space, which ... show a milder dependence on the initial value of ρ_a ...”.

The authors mention they used an aligning torque as assumption and implementation in the simulation to account for the emergence of path reuse. Yet, it is not clear what motivates the exclusive focus on torque, except for a post-hoc rationalization through experiments. This

might be the simplest hypothesis or the only possible hypothesis? It would be beneficial if the authors could expand upon their justifications of modeling the emergence as torque.

We thank the Reviewer for raising this point as it helps clarify an important aspect of our work. We used an aligning torque in our simulations to describe the emergence of path reuse following what we observed in our experiments with the active colloids (we refer to our reply to point 1 of Reviewer 1 for a more in-depth discussion on the physical origin of this torque). Briefly, the presence of obstacles in the proximity of an active particle can induce an asymmetric demixing of the critical mixture around the axis defined by its direction of motion. This produces a torque that reorients the particles towards the more demixed side (i.e., away from the obstacle, see Fig. R2 in our reply to Reviewer 1) when moving with the carbon cap at the front as in our experiments. This reorientation (and, hence, the presence of an aligning torque) continues until the particle is oriented in a direction where the demixing is symmetric again (for example, due to the presence of an open path or due to approaching two obstacles symmetrically) and the aligning torque vanishes.

This can be seen in Fig. R1 (see our reply to point 1 of Reviewer 1). At $t = 24$ s, a Janus particle approaches a group of relatively packed obstacles. On contact with an obstacle, the particle turns clockwise (between $t = 24$ s and $t = 34$ s) towards a void in the structure formed by the passive colloids instead of keeping on pushing in the same direction of approach. Later at $t = 47$ s, when it encounters a new block of relatively packed obstacles, a new reorientation event turns the particle counterclockwise based on its direction of approach, thus aligning the particle's motion to an open path, and enabling its reuse (between $t = 47$ s and $t = 58$ s).

In general, torques are commonly observed at the microscopic level. For example, aligning torques have been observed in the interaction of other active colloids with fixed physical boundaries (see, e.g., current references [39,40]) and with light gradients [see, e.g., Gomez Solano *et al.*, *Sci. Rep.* 7, 14891 (2017)]; torques have also been observed in the interaction of bacteria (such as *E. coli* cells) with boundaries and obstacles [see, e.g., our previous work: Makarchuk *et al.*, *Nature Communications* 10, 4110 (2019)]. Although, the presence of an aligning torque is the justification for the observed behaviour in our experiments, we agree with the Reviewer that similar alignment effects as those observed here could be observed independently of the specific physical mechanism that leads to particle's alignment to transient paths: for example, similar effects could also be observed in the presence of sensory perception, which could also be modelled with an effective torque in a coarse grained way.

To clarify this point, we have introduced the following changes to the manuscript:

- On page 3, we added: "...light absorption at the carbon cap simultaneously propels the Janus particles in the field of view with the more hydrophobic carbon-coated side at the front...".
- On pages 5-6, we included a detailed discussion of the physical mechanism behind the torque: "To shed light on the physical mechanism behind the reuse of paths in Fig. 1 and Supplementary Fig. 1, Figs. 3a-b show a close-up of the motion of a Janus particle in the background of passive particles. At $t = 24$ s (Fig. 3a), the particle approaches a block of relatively packed obstacles. On approaching one of them, it turns clockwise towards a void in the structure formed by the passive colloids instead of pushing ahead. At $t = 34$ s, the Janus particle approaches two obstacles head-on via their middle, thus pushing them out of the way and continuing its journey towards a pre-existing path (Fig. 3b). Interestingly, the particle does not align with this path until it encounters a new block of relatively packed obstacles ($t = 47$ s). At this point, a new reorientation event turns it counter-clockwise, thus aligning the particle's motion to the open path ($t = 58$ s) and allowing its reuse. To interpret these reorientation events, we need to consider how the presence of the obstacles affects the particle's self-propulsion mechanism. In their absence, light absorption at the particle's carbon cap induces the local demixing of the water-2,6-lutidine critical mixture, leading to the

formation of a lutidine-rich droplet around the more hydrophobic carbon side and of a smaller water-rich droplet around the more hydrophilic silica side [35, 42]. The formation of these droplets, which propels the particle [35, 42], is mostly symmetric around the axis defined by the direction of the particle's motion, and the particle's reorientation in a viscous medium is driven by its intrinsic rotational diffusion dynamics [35, 42]. If obstacles are present on one side, the local demixing is no longer symmetric around the direction of the particle's motion. This asymmetry induces a deterministic aligning torque that reorients the particle towards the more demixed side, i.e., eventually away from the obstacles [42]. As can be seen in Figs. 3a-b, this reorientation (and, hence, the presence of the torque) depends on the local configuration of obstacles and continues until the particle is aligned in a direction where the demixing around the caps is symmetric again and the aligning torque vanishes (for example, along an open path). To quantify the previous observation, we have calculated the angle δ at which the Janus colloids are deflected by their interaction with the obstacles (Fig. 3c). Positive and negative δ values indicate that the particles are deflected initially away from or towards the obstacles, respectively. When steering away from the obstacles (positive δ), we can then expect the particles to immediately align to openings in the background of passive particles, while deflections towards the obstacles (negative δ) would initially increase chances of collisions with the obstacles until the particle digs a new path or aligns to an existing one. The distribution of angles in Fig. 3c is peaked at positive δ values (as highlighted by the Gaussian fit centred at $\delta \approx 0.23$ rad), thus quantitatively confirming the stronger tendency for active particles to avoid obstacles and align to open paths already on approach."

- On page 6, we rephrased the introduction of the model as: "... we can therefore consider a simple particle-based model that includes an aligning torque ... which depends on the local configuration of obstacles (Methods)."
- We added a new figure to include the data in Figs. R1 and R2 (Fig. 3 in the new version of the manuscript).
- In the conclusions on page 9, we added the following discussion: "In our experiments, the physical mechanism behind the emergence of these population dynamics is the aligning torque acting on the Janus particles due to the asymmetric demixing of the water–2,6-lutidine critical mixture caused by the presence of the non-fixed obstacles. Other physical mechanisms (e.g., electrostatic interactions, phoretic interactions, hydrodynamic coupling [39] or, even, sensory perception [32, 49]) can be expected to lead to the emergence of a shared field memory in other systems, which could also be reproduced in a coarse-grained manner by introducing an effective aligning torque in their motion dynamics."
- We added Gomez Solano *et al.*, Sci. Rep. 7, 14891 (2017) as new reference [42].

The effects shown in Figs. 2b, 3b and the rate equation model seem to align well with other results from previous work, such as Hamann & Reina, Scalability in Computing and Robotics, 2021 also the rate equation model can be related to common approaches in swarm robotics. The authors may consider to draw the obvious connections also to related domains here.

We thank the Reviewer for pointing out the connection with the use of the rate equation model in robotics and swarm robotics in general. We have added a sentence when we first introduce the rate equation model (on page 7) to clarify that this approach is commonly used when studying the emergence of group dynamics and self-assembly in systems of multiple units, as much as in chemical kinetics and colloidal science as in swarm robotics: "Rate equations are indeed a powerful tool to understand the emergence of group dynamics and self-assembly in systems of multiple units, as much as in chemical kinetics [44] and colloidal science [12] as in swarm robotics [45, 46]." We have also added three new references ([44-46]), including the reference suggested by the Reviewer (now reference [46]).

Minor comment:

- It is not clear to me if all Janus particles are light activated at the same time or sequentially? A clarification might be useful. I understand that due to possibly very different timescales of activation and motion the sequential activation may not introduce any problems.

Our active particles are activated all at the same time by providing a constant homogenous illumination that expand over the entire field of view. We have now clarified this

- in the text on page 3, where we added: "...light absorption at the carbon cap simultaneously propels the Janus particles in the field of view...";
- and in the Methods on page 11, where we state: "The sample's field of view was illuminated at once with a green continuous-wave laser ($\lambda = 532$ nm) at a power density of $2.5 \mu\text{W} \mu\text{m}^{-2}$ to simultaneously propel the Janus particles due to light absorption at the carbon cap [35]."

Reviewer #3 (Remarks to the Author):

This is a well written paper on the interesting topic of stigmergic memory in a collective system of clueless (processing-less) individuals.

I think the paper is quite mature in its current form and I have only a few minor comments.

We thank the Reviewer for acknowledging the interest of our results and quality of our manuscript.

My main comment is regarding the "path revival lifetime" that is used in pages 5 and 6. The path revival function is explained already in the main text, but the "path revival lifetime" is not defined before the Methods section, which hinders the understanding of the discussions that follows (and the related figures). I think a clear description is necessary in the main text - maybe you can use the equation 3 and a short explanation on what it is.

Following the Reviewer's suggestion, we have now included an explanation of the path revival function in the main text on page 5, which reads: "The path reuse by the Janus particles can be quantified through the path revival function $1 - C_{aa}(\tau)$ [...]. **If we consider the particles' velocities to be Poisson distributed when a path is chosen, then this function should follow an exponential distribution for persistent particles of the form $1 - C_{aa}(\tau) = \exp(-\tau/\tau_{\rho_p})$ [41], where τ_{ρ_p} is the effective path revival lifetime, which we fit from the data (Supplementary Fig. 3). The shorter τ_{ρ_p} , the faster $1 - C_{aa}(\tau)$ decays (i.e., the faster $C_{aa}(\tau)$ increases to one), the sooner a region explored by a particle will be crossed by another particle, thus indicating a higher likelihood that a previously opened path will be reused by other active particles."**

Most of the links to figures S1, S2, S3 bring the reader to figures 1, 2, and 3 instead.

We thank the Reviewer for spotting this. The hyperlinks should now point to the correct supplementary figures.

Fig 3: What is survival time mentioned in the caption?

We thank the Reviewer for noticing this inconsistency in terminology. What we called "survival time" here is elsewhere referred to as the "path revival lifetime". We have now unified the terminology to "path revival lifetime".

P6: is there a repulsion between the active and passive particles that lead to the aligning torque, or what is the cause of the torque?

The reason for the aligning torque is an asymmetry in the demixing in the water–2,6-lutidine critical mixture around the Janus particles in the presence of the passive particles (a more in-depth discussion can be found in our reply to point 1 of Reviewer 1). Briefly, light absorption at the carbon cap (heated above the mixture's critical temperature) causes demixing of the critical mixture around the particle and the formation of a lutidine-rich droplet around the more hydrophobic carbon cap and of a smaller water-rich droplet around the more hydrophilic silica side [see, e.g., Buttinoni *et al.*, J. Phys.: Condens. Matter 24, 284129 (2012)]. The formation of these droplets causes self-propulsion because of symmetry breaking around the particle. In the absence of obstacles, these droplets are mostly symmetric around the direction of self-propulsion [see, e.g., Buttinoni *et al.*, J. Phys.: Condens. Matter 24, 284129 (2012); Gomez Solano *et al.*, Sci. Rep. 7, 14891 (2017)] and the particle's re-orientational dynamics are purely due to its rotational diffusion [see, e.g., Gomez Solano *et al.*, Sci. Rep. 7, 14891 (2017)]. In the presence of obstacles, demixing is no longer symmetric around each cap as limited on the obstacle side. This asymmetry induces a torque that reorients the particles towards the more

demixed side (see Fig. R2 in our reply to point 1 of Reviewer 1) when moving with the carbon cap at the front as in our experiments. This reorientation (and, hence, the presence of a torque) continues until the particle is oriented in a direction where the demixing is symmetric again (e.g., by aligning to an open path) and the torque vanishes. This can be seen in the time sequence of Fig. R1 (see our reply to point 1 of Reviewer 1), for example.

To clarify this point, we have introduced the following changes to the manuscript:

- On page 3, we added: "...light absorption at the carbon cap simultaneously propels the Janus particles in the field of view with the more hydrophobic carbon-coated side at the front...".
- On pages 5-6, we included a detailed discussion of the physical mechanism behind the torque: "To shed light on the physical mechanism behind the reuse of paths in Fig. 1 and Supplementary Fig. 1, Figs. 3a-b show a close-up of the motion of a Janus particle in the background of passive particles. At $t = 24$ s (Fig. 3a), the particle approaches a block of relatively packed obstacles. On approaching one of them, it turns clockwise towards a void in the structure formed by the passive colloids instead of pushing ahead. At $t = 34$ s, the Janus particle approaches two obstacles head-on via their middle, thus pushing them out of the way and continuing its journey towards a pre-existing path (Fig. 3b). Interestingly, the particle does not align with this path until it encounters a new block of relatively packed obstacles ($t = 47$ s). At this point, a new reorientation event turns it counter-clockwise, thus aligning the particle's motion to the open path ($t = 58$ s) and allowing its reuse. To interpret these reorientation events, we need to consider how the presence of the obstacles affects the particle's self-propulsion mechanism. In their absence, light absorption at the particle's carbon cap induces the local demixing of the water–2,6-lutidine critical mixture, leading to the formation of a lutidine-rich droplet around the more hydrophobic carbon side and of a smaller water-rich droplet around the more hydrophilic silica side [35, 42]. The formation of these droplets, which propels the particle [35, 42], is mostly symmetric around the axis defined by the direction of the particle's motion, and the particle's reorientation in a viscous medium is driven by its intrinsic rotational diffusion dynamics [35, 42]. If obstacles are present on one side, the local demixing is no longer symmetric around the direction of the particle's motion. This asymmetry induces a deterministic aligning torque that reorients the particle towards the more demixed side, i.e., eventually away from the obstacles [42]. As can be seen in Figs. 3a-b, this reorientation (and, hence, the presence of the torque) depends on the local configuration of obstacles and continues until the particle is aligned in a direction where the demixing around the caps is symmetric again and the aligning torque vanishes (for example, along an open path). To quantify the previous observation, we have calculated the angle δ at which the Janus colloids are deflected by their interaction with the obstacles (Fig. 3c). Positive and negative δ values indicate that the particles are deflected initially away from or towards the obstacles, respectively. When steering away from the obstacles (positive δ), we can then expect the particles to immediately align to openings in the background of passive particles, while deflections towards the obstacles (negative δ) would initially increase chances of collisions with the obstacles until the particle digs a new path or aligns to an existing one. The distribution of angles in Fig. 3c is peaked at positive δ values (as highlighted by the Gaussian fit centred at $\delta \approx 0.23$ rad), thus quantitatively confirming the stronger tendency for active particles to avoid obstacles and align to open paths already on approach."
- On page 6, we rephrased the introduction of the model as: "... we can therefore consider a simple particle-based model that includes an aligning torque ... which depends on the local configuration of obstacles (Methods)."
- We added a new figure to include the data in Figs. R1 and R2 (Fig. 3 in the new version of the manuscript).
- We added Gomez Solano *et al.*, Sci. Rep. 7, 14891 (2017) as new reference [42].

REVIEWERS' COMMENTS

Reviewer #1 (Remarks to the Author):

The revised manuscript and reply are all well written. I understand that the system under consideration is in the Stokes regime. In such a case, the analysis, modeling and discussions are all justified.

Therefore I recommend to publish this paper Nature Communications.

Reviewer #2 (Remarks to the Author):

In this revision, the authors have addressed all of the comments raised by the reviewers in the initial round of reviews. I believe that these revisions, and especially the expanded discussion about the role of stigmergy and the physical causes of the aligning torque, have improved the quality of the manuscript and no further open issues remain.

Reviewer #3 (Remarks to the Author):

Thank you for the changes. I particularly enjoyed reading the added clarifying discussion on the mechanisms behind the torque.

I think the paper is interesting and well-written.

Manuscript NCOMMS-23-25939A
Authors' Response to Final Reviewers' Comments

We are grateful to all Reviewers for carefully re-reading our manuscript and for their final positive assessment.

Reviewer #1 (Remarks to the Author):

The revised manuscript and reply are all well written. I understand that the system under consideration is in the Stokes regime. In such a case, the analysis, modeling and discussions are all justified. Therefore I recommend to publish this paper Nature Communications.

We thank the Reviewer for their recommendation to publish our work in *Nature Communications*.

Reviewer #2 (Remarks to the Author):

In this revision, the authors have addressed all of the comments raised by the reviewers in the initial round of reviews. I believe that these revisions, and especially the expanded discussion about the role of stigmergy and the physical causes of the aligning torque, have improved the quality of the manuscript and no further open issues remain.

We thank the Reviewer for their final assessment. We are glad that all comments from the previous round have been addressed.

Reviewer #3 (Remarks to the Author):

Thank you for the changes. I particularly enjoyed reading the added clarifying discussion on the mechanisms behind the torque. I think the paper is interesting and well-written.

We thank the Reviewer for acknowledging the interest and quality of our paper.